# Meta Flow Matching: Integrating Vector Fields on the Wasserstein Manifold

**Lazar Atanackovic**[1,2*]   **Xi Zhang**[3,4*]   **Brandon Amos**[5]   **Mathieu Blanchette**[3,4]
**Leo J. Lee**[1,2]   **Yoshua Bengio**[3,6,7]   **Alexander Tong**[3,6]   **Kirill Neklyudov**[3,6]
[1]University of Toronto   [2]Vector Institute   [3]Mila - Quebec AI Institute
[4]McGill University   [5]Meta   [6]Université de Montréal   [7]CIFAR Fellow

## Abstract

Numerous biological and physical processes can be modeled as systems of interacting entities evolving continuously over time, e.g. the dynamics of communicating cells or physical particles. Learning the dynamics of such systems is essential for predicting the temporal evolution of populations across novel samples and unseen environments. Flow-based models allow for learning these dynamics at the population level — they model the evolution of the entire distribution of samples. However, current flow-based models are limited to a single initial population and a set of predefined conditions which describe different dynamics. We argue that multiple processes in natural sciences have to be represented as vector fields on the Wasserstein manifold of probability densities. That is, the change of the population at any moment in time depends on the population itself due to the interactions between samples. In particular, this is crucial for personalized medicine where the development of diseases and their respective treatment response depend on the microenvironment of cells specific to each patient. We propose *Meta Flow Matching* (MFM), a practical approach to integrate along these vector fields on the Wasserstein manifold by amortizing the flow model over the initial populations. Namely, we embed the population of samples using a Graph Neural Network (GNN) and use these embeddings to train a Flow Matching model. This gives MFM the ability to generalize over the initial distributions, unlike previously proposed methods. We demonstrate the ability of MFM to improve the prediction of individual treatment responses on a large-scale multi-patient single-cell drug screen dataset.

## 1 Introduction

Understanding the dynamics of many-body problems is a focal challenge across the natural sciences. In the field of cell biology, a central focus is the understanding of the dynamic processes that cells undergo in response to their environment, and in particular their response and interaction with other cells. Cells communicate with one other in close proximity using *cell signaling*, exerting influence over each other's trajectories (Armingol et al., 2020; Goodenough and Paul, 2009). This signaling presents an obstacle for modeling due to the complex nature of intercellular regulation, but is essential for understanding and eventually controlling cell dynamics during development (Gulati et al., 2020; Rizvi et al., 2017), in diseased states (Molè et al., 2021; Binnewies et al., 2018; Zeng and Dai, 2019; Chung et al., 2017), and in response to perturbations (Ji et al., 2021; Peidli et al., 2024).

The exponential decrease of sequencing costs and advances in microfluidics has enabled the rapid advancement of single-cell sequencing and related technologies over the past decade (Svensson et al., 2018). While single-cell sequencing has been used to great effect to understand the heterogeneity in cell systems, it is also destructive, making longitudinal measurements extremely difficult. Hence, learning dynamical system models of cells while also capturing their inherent heterogeneity and stochasticity of cellular systems remains a central challenge in biology.

Instead, most existing approaches model cell dynamics at the population level (Hashimoto et al., 2016; Weinreb et al., 2018; Schiebinger et al., 2019; Tong et al., 2020; Neklyudov et al., 2022; Bunne et al., 2023). These approaches involve the formalisms of optimal transport (Villani, 2009; Peyré and Cuturi, 2019), diffusion (De Bortoli et al., 2021), or normalizing flows (Lipman et al., 2022), which

---

*Joint first authorship. Correspondence to: `l.atanackovic@mail.utoronto.ca`
Our code is available at: `https://github.com/lazaratan/meta-flow-matching`

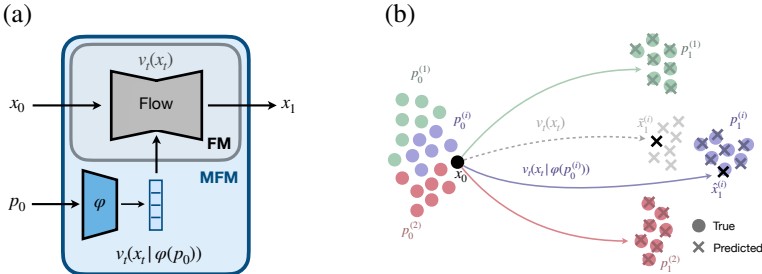

Figure 1: **Illustration of *Meta Flow Matching* (MFM, Eq. (17)).** (a) Comparison between *Flow Matching* (FM, Eq. (6)) and MFM. (b) Depiction of differences between MFM and FM generated predictions. Given a point $x_t$, a vector field (flow) model trained with MFM can generate different points $\hat{x}_1$ for different initial distributions $p_0$ (represented by red, green, and purple). FM trained models can only predict an aggregate response over populations (shown in gray). FM at best can incorporate known (seen) conditional information available in the training data, denoted as *Conditional Generative Flow Matching* (CGFM, Eq. (9)). In contrast, MFM jointly learns a population embedding model $\varphi(p_0)$ and a vector field $v_t$, allowing generalization to unseen populations.

learn a map between empirical measures. While these methods are able to model the dynamics of the population, they are fundamentally limited in that they model the evolution of cells as independent particles evolving according to the learned model of a global vector field. Furthermore, these models can be trained to match any given set of measures, but they are restricted to modeling of a single population and can at best condition on a number of different dynamics that are in the training data.

**We propose *Meta Flow Matching* (MFM) — the amortization of the Flow Matching generative modeling framework** (Lipman et al., 2022) **over the input measures**, which (i) takes into account the interaction of particles (instead of modelling them independently) and (ii) allows for generalization of the learned model to previously unseen input populations. In practice, our method can be used to predict the time-evolution of distributions from a given dataset of the time-evolved examples. Namely, we assume that the collected data undergoes a universal developmental process, which depends only on the population itself as in the setting of the interacting particles or communicating cells. Under this assumption, we learn the vector field model that takes samples from the initial distribution as input and defines the push-forward map on the sample-space that maps the initial distribution to the final distribution (see Fig. 1).

We showcase the utility of our approach on two applications. To illustrate the intuition of the proposed method, we first test MFM on a synthetic task of "letter denoising". We show that MFM is able to generalize the denoising process to unseen letter silhouettes, whereas the standard FM approach cannot. Next, we explore how MFM can be applied to model single-cell perturbation data (Ji et al., 2021; Peidli et al., 2024). We evaluate MFM on predicting the response of patient-derived cells to chemotherapy treatments in a recently published large scale single-cell drug screening dataset where there are known to be patient-specific responses (Ramos Zapatero et al., 2023).[1] This is a challenging task due to the inherent variation that exist across patients, the variability induced by different treatments, and variation due to local cell compositions. Addressing this problem can lead to better prediction of tumor growth and therapeutic response to cancer treatment. We demonstrate that MFM can successfully predict the development of cell populations on replicated experiments, and most importantly, that it generalizes to previously unseen patients, thus, capturing the patient-specific response to the treatment.

## 2 BACKGROUND

### 2.1 GENERATIVE MODELING VIA FLOW MATCHING

Flow Matching is an approach to generative modeling recently proposed independently in different works: Rectified Flows (Liu et al., 2022b), Flow Matching (Lipman et al., 2022), Stochastic Interpolants (Albergo and Vanden-Eijnden, 2022). A flow is a continuous interpolation between densities $p_0(x_0)$ and $p_1(x_1)$ in the sample space. That is, the sample from the intermediate density $p_t(x)$ is produced as follows

$$x_t = f_t(x_0, x_1), \quad (x_0, x_1) \sim \pi(x_0, x_1), \tag{1}$$

$$\text{where} \int dx_1 \, \pi(x_0, x_1) = p_0(x_0), \quad \int dx_0 \, \pi(x_0, x_1) = p_1(x_1), \tag{2}$$

---

[1]This dataset includes more than 25 million cells collected for ten patients over 2500 different experimental/treatment conditions.

where $f_t$ is the time-continuous interpolating function such that $f_{t=0}(x_0, x_1) = x_0$ and $f_{t=1}(x_0, x_1) = x_1$ (e.g. linearly between $x_0$ and $x_1$ with $f_t(x_0, x_1) = (1-t) \cdot x_0 + t \cdot x_1$); $\pi(x_0, x_1)$ is the density of the joint distribution, which is usually taken as a distribution of independent random variables $\pi(x_0, x_1) = p_0(x_0)p_1(x_1)$, but can also be generalized to formulate the optimal transport problems (Pooladian et al., 2023; Tong et al., 2024a). The corresponding density can be defined then as the following expectation over the interpolating samples

$$p_t(x) = \int dx_0 dx_1 \, \pi(x_0, x_1)\delta(x - f_t(x_0, x_1)) \,. \tag{3}$$

The essential part of Flow Matching is the continuity equation that describes the change of this density through the vector field on the state space, which admits vector field $v_t^*(x)$ as a solution

$$\frac{\partial p_t(x)}{\partial t} = -\langle \nabla_x, p_t(x)v_t^*(x)\rangle \,, \quad v_t^*(\xi) = \frac{1}{p_t(\xi)}\mathbb{E}_{\pi(x_0, x_1)}\left[\delta(f_t(x_0, x_1) - \xi)\frac{\partial f_t(x_0, x_1)}{\partial t}\right] \,. \tag{4}$$

The expectation for $v_t^*$ is intractable, but we can model $v_t(x; \omega)$ to approximate $v_t^*$ so that we can efficiently generate new samples. Relying on Eq. (4), one can derive the tractable objective for learning $v_t^*(x)$, i.e.

$$\mathcal{L}_{\text{FM}}(\omega) = \int_0^1 dt \, \mathbb{E}_{p_t(x)}\|v_t^*(x) - v_t(x; \omega)\|^2 \tag{5}$$

$$= \mathbb{E}_{\pi(x_0, x_1)} \int_0^1 dt \, \left\|\frac{\partial}{\partial t} f_t(x_0, x_1) - v_t(f_t(x_0, x_1); \omega)\right\|^2 + \text{constant} \,. \tag{6}$$

Finally, the vector field $v_t(\xi, \omega) \approx v_t^*(\xi)$ defines the push-forward density that approximately matches $p_{t=1}$, i.e. $T_{\#}p_0 \approx p_{t=1}$, where $T$ is the flow corresponding to vector field $v_t(\cdot, \omega)$ with parameters $\omega$.

## 2.2 Conditional Generative Modeling via Flow Matching

Conditional image generation is one of the most common applications of generative models nowadays; it includes conditioning on the text prompts (Saharia et al., 2022b; Rombach et al., 2022) as well as conditioning on other images (Saharia et al., 2022a). To learn the conditional generative process with diffusion models, one merely has to pass the conditional variable (sampled jointly with the data point) as an additional input to the parametric model of the vector field. The same applies for the Flow Matching framework.

Conditional Generative Modeling via Flow Matching is independently introduced in several works (Zheng et al., 2023; Dao et al., 2023; Isobe et al., 2024) and it operates as follows. Consider a family of time-continuous densities $p_t(x_t \mid c)$, which corresponds to the distribution of the following random variable

$$x_t = f_t(x_0, x_1), \quad (x_0, x_1) \sim \pi(x_0, x_1 \mid c) \,. \tag{7}$$

For every $c$, the density $p_t(x_t \mid c)$ follows the continuity equation with the following vector field

$$v_t^*(\xi \mid c) = \frac{1}{p_t(\xi \mid c)}\mathbb{E}_{\pi(x_0, x_1)}\delta(f_t(x_0, x_1) - \xi)\frac{\partial f_t(x_0, x_1)}{\partial t} \,, \tag{8}$$

which depends on $c$. Thus, the training objective of the conditional model becomes

$$\mathcal{L}_{CGFM}(\omega) = \mathbb{E}_{p(c)}\mathbb{E}_{\pi(x_0, x_1 \mid c)}\int_0^1 dt \, \left\|\frac{\partial}{\partial t} f_t(x_0, x_1) - v_t(f_t(x_0, x_1) \mid c; \omega)\right\|^2 \,, \tag{9}$$

where, compared to the original Flow Matching formulation, we first have to sample $c$, then produce the samples from $p_t(x_t \mid c)$ and pass $c$ as input to the parametric model of the vector field.

## 2.3 Modeling Process in Natural Sciences as Vector Fields on the Wasserstein Manifold

We argue that numerous biological and physical processes cannot be modeled via the vector field propagating the population samples independently. Thus, we propose to model these processes as families of conditional vector fields where we amortize the conditional variable by embedding the population via a Graph Neural Network (GNN).

To provide the reader with the necessary intuition, we are going to use the geometric formalism developed by Otto (2001) (see the complete discussion in Appendix A). That is, time-dependent

**Figure 2: Illustration of flow matching methods on the 2-Wasserstein manifold, $\mathcal{P}_2(\mathcal{X})$, depicted as a two-dimensional sphere.** *Flow Matching* learns the tangent vectors to a single curve on the manifold. *Conditional* generation corresponds to learning a finite set of curves on the manifold, e.g. classes $c_1$ and $c_2$ on the plot. *Meta Flow Matching* learns to integrate a vector field on $\mathcal{P}_2(\mathcal{X})$, i.e. for every starting density $p_0$, MFM defines a push-forward measure that integrates along the underlying vector field.

densities $p_t(x_t)$ define absolutely-continuous curves on the 2-Wasserstein space of distributions $\mathcal{P}_2(\mathcal{X})$ (see (Ambrosio et al., 2008), Chapter 8). The tangent space of this manifold is defined by the gradient flows $\mathcal{S}_t = \{\nabla s_t \mid s_t : \mathcal{X} \to \mathbb{R}\}$ on the state space $\mathcal{X}$, to be precise, its $L^2(\mu; \mathcal{X})$ closure $\forall \mu \in \mathcal{P}_2(\mathcal{X})$ (see (Ambrosio et al., 2008), eq. (8.0.2)). In the Flow Matching context, we are going to refer to the tangent vectors as vector fields since one can always project the vector field onto the tangent space by parameterizing it as a gradient flow (Neklyudov et al., 2022).

Under the geometric formalism of the 2-Wasserstein manifold, Flow Matching can be considered as learning the tangent vectors $v_t(\cdot)$ along the density curve $p_t(x_t)$ defined by the sampling process in Eq. (2) (see the left panel in Fig. 2). Furthermore, the conditional generation processes $p_t(x_t \mid c)$ would be represented as a finite set of curves if $c$ is discrete (e.g. class-conditional generation of images) or as a family of curves if $c$ is continuous (see the middle panel in Fig. 2).

Finally, one can define a vector field on the 2-Wasserstein manifold via the continuity equation with the vector field $v_t(x, p_t(x))$ on the state space $\mathcal{X}$ that depends on the current density $p_t(x)$ or its derivatives. Below we give two examples of processes defined as vector fields on the 2-Wasserstein manifold.

**Example 1** (Mean-field limit of interacting particles). *Consider a system of interacting particles, where the velocity of the particle at point $x$ interacting with the particle at point $y$ is defined as $k(x, y) : \mathbb{R}^d \times \mathbb{R}^d \to \mathbb{R}^d$. In the limit of the infinite number of particles one can describe their state using the density function $p_t(x)$. Then the change of the density is described by the following continuity equation*

$$\frac{dx}{dt} = \mathbb{E}_{p_t(y)} k(x, y), \quad \frac{\partial p_t(x)}{\partial t} = -\left\langle \nabla_x, p_t(x) \mathbb{E}_{p_t(y)} k(x, y) \right\rangle, \tag{10}$$

*which is the first-order analog of the Vlasov equation (Jabin and Wang, 2016).*

**Example 2** (Diffusion). *Even when the physical particles evolve independently in nature, the deterministic vector field model might be dependent on the current density of the population. For instance, for the diffusion process, the change of the density is described by the Fokker-Planck equation, which results in the density-dependent vector field when written as a continuity equation, i.e.*

$$\frac{\partial p_t(x)}{\partial t} = \frac{1}{2} \Delta_x p_t(x) = -\left\langle \nabla_x, p_t(x) \left( -\frac{1}{2} \nabla_x \log p_t(x) \right) \right\rangle \implies \frac{dx}{dt} = -\frac{1}{2} \nabla_x \log p_t(x). \tag{11}$$

Motivated by the examples above, we argue that using the information about the current or the initial density is crucial for the modeling of time-evolution of densities in natural processes, to capture this type of dependency one can model the change of the density as the following Cauchy problem

$$\frac{\partial p_t(x)}{\partial t} = -\left\langle \nabla_x, p_t(x) v_t(x, p_t) \right\rangle, \quad p_{t=0}(x) = p_0(x), \tag{12}$$

where the state-space vector field $v_t(x, p_t)$ depends on the density $p_t$.

The dependency might vary across models, e.g. in Example 1 the vector field can be modeled as an application of a kernel to the density function, while in Example 2 the vector field depends only on the local value of the density and its derivative.

## 3 META FLOW MATCHING

In this paper, we propose the amortization of the Flow Matching framework over the marginal distributions. Our model is based on the outstanding ability of the Flow Matching framework to learn the push-forward map for any joint distribution $\pi(x_0, x_1)$ given empirically. For the given joint $\pi(x_0, x_1)$, we denote the solution of the Flow Matching optimization problem as follows

$$v_t^*(\cdot, \pi) = \arg\min_{v_t} \mathcal{L}_{GFM}(v_t(\cdot), \pi(x_0, x_1)). \tag{13}$$

Analogous to amortized optimization (Chen et al., 2022; Amos et al., 2023), we aim to learn the model that outputs the solution of Eq. (13) based on the input data sampled from $\pi$, i.e.

$$v_t(\cdot, \varphi(\pi)) = v_t^*(\cdot, \pi), \tag{14}$$

where $\varphi(\pi)$ is the embedding model of $\pi$ and the joint density $\pi(\cdot \,|\, c)$ is generated using some unknown measure of the conditional variables $c \sim p(c)$.

### 3.1 INTEGRATING VECTOR FIELDS ON THE WASSERSTEIN MANIFOLD VIA META FLOW MATCHING

Consider the dataset of joint populations $\mathcal{D} = \{(\pi(x_0, x_1 \,|\, i))\}_i$, where, to simplify the notation, we associate every $i$-th population with its density $\pi(\cdot \,|\, i)$ and the conditioning variable here is the index of this population in the dataset. We make the following assumptions regarding the ground truth sampling process (i) we assume that the starting marginals $p_0(x_0 \,|\, i) = \int dx_1\, \pi(x_0, x_1 \,|\, i)$ are sampled from some unknown distribution that can be parameterized with a large enough number of parameters (ii) the endpoint marginals $p_1(x_1 \,|\, i) = \int dx_0\, \pi(x_0, x_1 \,|\, i)$ are obtained as push-forward densities solving the Cauchy problem in Eq. (12), (iii) there exists unique solution to this Cauchy problem.

One can learn a joint model of all the processes from the dataset $\mathcal{D}$ using the conditional version of the Flow Matching algorithm (see Section 2.2) where the population index $i$ plays the role of the conditional variable. However, obviously, such a model will not generalize beyond the considered data $\mathcal{D}$ and unseen indices $i$. We illustrate this empirically in Section 5.

To be able to generalize to previously unseen populations, we propose learning the density-dependent vector field motivated by Eq. (12). That is, we propose to use an embedding function $\varphi : \mathcal{P}_2(\mathcal{X}) \to \mathbb{R}^m$ to embed the starting marginal density $p_0$, which we then input into the vector field model and minimize the following objective over $\omega$

$$\mathcal{L}_{\mathrm{MFM}}(\omega; \varphi) = \mathbb{E}_{i \sim \mathcal{D}} \mathbb{E}_{\pi(x_0, x_1 \,|\, i)} \int_0^1 dt \left\| \frac{\partial}{\partial t} f_t(x_0, x_1) - v_t(f_t(x_0, x_1) \,|\, \varphi(p_0); \omega) \right\|^2. \tag{15}$$

Note that the initial density $p_0$ is enough to predict the push-forward density $p_1$ since the Cauchy problem for Eq. (12) has a unique solution. The embedding function $\varphi(p_0)$ can take different forms, e.g. it can be the density value $\varphi(p_0) = p_0(\cdot)$, which is then used inside the vector field model to evaluate at the current point (analogous to Example 2); a kernel density estimator (analogous to Example 1); or a parametric model taking the samples from this density as an input.

> **Proposition 1.** *Meta Flow Matching recovers the Conditional Generation via Flow Matching when the conditional dependence of the marginals $p_0(x_0 \,|\, c) = \int dx_1 \pi(x_0, x_1 \,|\, c)$ and $p_1(x_1 \,|\, c) = \int dx_0 \pi(x_0, x_1 \,|\, c)$ and the distribution $p(c)$ are known, i.e. there exist $\varphi : \mathcal{P}_2(\mathcal{X}) \to \mathbb{R}^m$ such that $\mathcal{L}_{MFM}(\omega) = \mathcal{L}_{CGFM}(\omega)$.*
>
> *Proof.* Indeed, sampling from the dataset $i \sim \mathcal{D}$ becomes sampling of the conditional variable $c \sim p(c)$ and the embedding function becomes $\varphi(p_0(\cdot \,|\, c)) = c$. □

Furthermore, for the parametric family of the embedding models $\varphi(p_t, \theta)$, we show that the parameters $\theta$ can be estimated by minimizing the objective in Eq. (15) in the joint optimization with the vector field parameters $\omega$. We formalize this statement in the following theorem.

---

**Algorithm 1:** Meta Flow Matching (training)

---

**Input :** dataset of populations $\{(\pi(x_0, x_1 \mid i), c^i)\}_{i=1}^N$ and treatments $c^i$, and parametric models
for the velocity, $v_t(\cdot; \omega)$, and population embedding $\varphi(\cdot; \theta)$.

**for** *training iterations* **do**

$\quad i \sim \mathcal{U}_{\{1,N\}}(i)$ // sample batch of $n$ populations ids

$\quad (x_0^j, x_1^j, t^j) \sim \pi(x_0, x_1 \mid i)\mathcal{U}_{[0,1]}(t)$ // sample $N_i$ particles for every population $i$

$\quad f_t(x_0^j, x_1^j) \leftarrow (1 - t^j)x_0^j + t^j x_1^j$

$\quad h^i(\theta) \leftarrow \varphi\Big(\{x_0^j\}_{j=1}^{N_i}; \theta\Big)$ // embed population $\{x_0^j\}_{j=1}^{N_i}$. For **CGFM** $h \leftarrow i$, **FM** $h \leftarrow \emptyset$.

$\quad \mathcal{L}_{\text{MFM}}(\omega, \theta) \leftarrow \frac{1}{n} \sum_i \frac{1}{n_i} \sum_j \left\| \frac{d}{dt} f_t(x_0^j, x_1^j) - v_{t^j}\Big(f_t(x_0^j, x_1^j) \mid h^i(\theta), c^i; \omega\Big) \right\|^2$

$\quad \omega' \leftarrow \text{Update}(\omega, \nabla_\omega \mathcal{L}_{\text{MFM}}(\omega, \theta))$ // evaluate new parameters of the flow model

$\quad \theta' \leftarrow \text{Update}(\theta, \nabla_\theta \mathcal{L}_{\text{MFM}}(\omega, \theta))$ // evaluate new parameters of the embedding model

$\quad \omega \leftarrow \omega', \ \theta \leftarrow \theta'$ // update both models

**return** $v_t(\cdot; \omega^*), \varphi(\cdot; \theta^*)$

---

**Theorem 1.** *Consider a dataset of populations $\mathcal{D} = \{(\pi(x_0, x_1 \mid i))\}_i$ generated from some unknown conditional model $\pi(x_0, x_1 \mid c)p(c)$. Then the following objective*

$$\mathcal{L}(\omega, \theta) = \mathbb{E}_{p(c)} \int_0^1 dt \, \mathbb{E}_{p_t(x_t \mid c)} \|v_t^*(x_t \mid c) - v_t(x_t \mid \varphi(p_0, \theta), \omega)\|^2 \quad (16)$$

*is equivalent to the Meta Flow Matching objective*

$$\mathcal{L}_{\text{MFM}}(\omega, \theta) = \mathbb{E}_{i \sim \mathcal{D}} \mathbb{E}_{\pi(x_0, x_1 \mid i)} \int_0^1 dt \left\| \frac{\partial}{\partial t} f_t(x_0, x_1) - v_t(f_t(x_0, x_1) \mid \varphi(p_0, \theta); \omega) \right\|^2 \quad (17)$$

*up to an additive constant.*

*Proof.* We postpone the proof to Appendix B. $\qquad\square$

## 3.2 Learning Population Embeddings via Graph Neural Networks (GNNs)

In many applications, the populations $\mathcal{D} = \{(\pi(x_0, x_1 \mid i))\}_{i=1}^N$ are given as empirical distributions, i.e. they are represented as samples from some unknown density $\pi$

$$\{(x_0^j, x_1^j)\}_{j=1}^{N_i}, \quad (x_0^j, x_1^j) \sim \pi(x_0, x_1 \mid i), \quad (18)$$

where $N_i$ is the size of the $i$-th population. For instance, for the diffusion process considered in Example 2, the samples from

---

**Algorithm 2:** Meta Flow Matching (sampling)

---

**Input :** initial population $\{x_0^j\}_{j=1}^{N'}$, treatment
condition $c^i$, models $v_t(\cdot; \omega^*)$ and $\varphi(\cdot; \theta^*)$.

$h = \varphi\Big(\{x_0^j\}_{j=1}^{N'}; \theta\Big)$ // embed the population

$x_1^j = \int_0^1 v_t(x_t^j \mid h, c^i; \omega)dt + x_0^j$ // ODE solver

**return** *predicted population* $\{x_1^j\}_{j=1}^{N'}$

---

$\pi(x_0, x_1 \mid i)$ can be generated by generating some marginal $p_1(x_1 \mid i)$ and then adding the Gaussian random variable to the samples $x_1^j$. We use this model in our synthetic experiments in Section 5.1.

Since the only available information about the populations is samples, we propose learning the embedding of populations via a parametric model $\varphi(p_0, \theta)$, i.e.

$$\varphi(p_0, \theta) = \varphi\Big(\{x_0^j\}_{j=1}^{N_i}, \theta\Big), \quad (x_0^j, x_1^j) \sim \pi(x_0, x_1 \mid i). \quad (19)$$

For this purpose, we employ GNNs, which recently have been successfully applied for simulation of complicated many-body problems in physics (Sanchez-Gonzalez et al., 2020). To embed a population $\{x_0^j\}_{j=1}^{N_i}$, we create a k-nearest neighbour graph $G_i$ based on the metric in the state-space $\mathcal{X}$, input it into a GNN, which consists of several message-passing iterations (Gilmer et al., 2017) and the final average-pooling across nodes to produce the embedding vector. Finally, we update the parameters of

the GNN jointly with the parameters of the vector field to minimize the loss function in Eq. (17). We show pseudo-code for training in Algorithm 1 and sampling in Algorithm 2.

## 4 RELATED WORK

***Meta – amortizing learning over distributions***     The meta-learning of probability measures was previously studied by Amos et al. (2022) where they demonstrated that the prediction of the optimal transport paths can be efficiently amortized over the input marginal measures. The main difference with our approach is that we are trying to learn the push-forward map without embedding the second marginal and without restricting ourselves to Input-Convex Neural Network (ICNN) by Amos et al. (2017) and optimal transport maps. MFM is *meta* in the same sense of amortizing optimal transport (or in our case flow) problems over multiple input distributions. Hence, we follow the naming convention of *Meta Optimal Transport* (Amos et al., 2022). This is different but related to how *meta* is used in the meta learning setting, which amortizes over learning problems (Hospedales et al., 2021; Achille et al., 2019).

**Generative modeling for single cells**     Single cell data has expanded to encompass multiple modalities of data profiling cell state and activities (Frangieh et al., 2021; Bunne et al., 2023). Single-cell data presents multiple challenges in terms of noise, non-time resolved, and high dimension, and generative models have been used to counter those problems. Autoencoder has been used to embed and extrapolate data Out Of Distribution (OOD) with its latent state dimension (Lotfollahi et al., 2019; Lopez et al., 2018; Hetzel et al., 2022). Orthogonal non-negative matrix factorization (oNMF) has also been used for dimensionality reduction combined with mixture models for cell state prediction (Chen et al., 2020). Other approaches have tried to use Flow Matching (FM) (Tong et al., 2024b;a; Neklyudov et al., 2023) or similar approaches such as the Monge gap (Uscidda and Cuturi, 2023) to predict cell trajectories. Currently, the state of the art method uses the principle of Optimal Transport (OT) to predict cell trajectories (Makkuva et al., 2020; Bunne et al., 2023). These methods are based on input convex neural network (ICNN) architectures and can generalize out of distribution to a new cells. As of this time, our method is the only method that takes inter-cellular interactions into account and learns embeddings of entire cell populations to generalize across unseen distributions.

**Generative modeling for physical processes**     The closest approach to ours is the prediction of the many-body interactions in physics (Sanchez-Gonzalez et al., 2020) via GNNs. However, the problem there is very different since these models use the information about the individual trajectories of samples, which are not available for the single-cell prediction. Liu et al. (2022a; 2024) consider a generalized form of Schrödinger bridges also with interacting terms, but do not consider the generalization to unseen distributions. Neklyudov et al. (2022) consider learning the vector field for any continuous time-evolution of a probability measure, however, their method is restricted to single curves and do not consider generalization to unseen data. Campbell et al. (2024) use input and space independent conditions for applications in co-protein design, but similarly, their method cannot generalize to novel distributions. Finally, the weather/climate forecast models generating the next state conditioned on the previous one (Price et al., 2023; Verma et al., 2024) are similar approaches to ours but operating on a much finer time resolution.

## 5 EXPERIMENTS

To show the effectiveness of MFM to generalize under previously unseen populations for the population prediction task, we consider two experimental settings. (i) A synthetic experiment with well defined coupled populations, and (ii) experiments on a publicly available single-cell dataset consisting of populations from patient dependent treatment response trials. We parameterize all vector field models $v_t(\cdot \mid \varphi(p_0); \omega)$ using a Multi-Layer Perceptron (MLP). For MFM, we additionally parameterize $\varphi(p_t; \theta, k)$ using a Graph Convolutional Network (GCN) with a $k$-nearest neighbor graph edge pooling layer. We include details regarding model hyperparameters, training/optimization, and implementation in Appendix D and Appendix D.2. We report results over 3 random seeds.

### 5.1 SYNTHETIC EXPERIMENT

**Synthetic data.**     We curate a synthetic dataset of the joint distributions $\{(p_0(x_0, \mid i), p_1(x_1 \mid i))\}_{i=1}^{N}$ by simulating a diffusion process applied to a set of pre-defined target distributions $p_1(x_1 \mid i)$ for $i = 1, \ldots, N$. To get a paired population $p_0(x_0 \mid i)$ we simulate the forward diffusion process without drift $x_0 \sim \mathcal{N}(x_1, \sigma)$. After this setup, for reasonable values of $\sigma$, we assume that one can reverse the

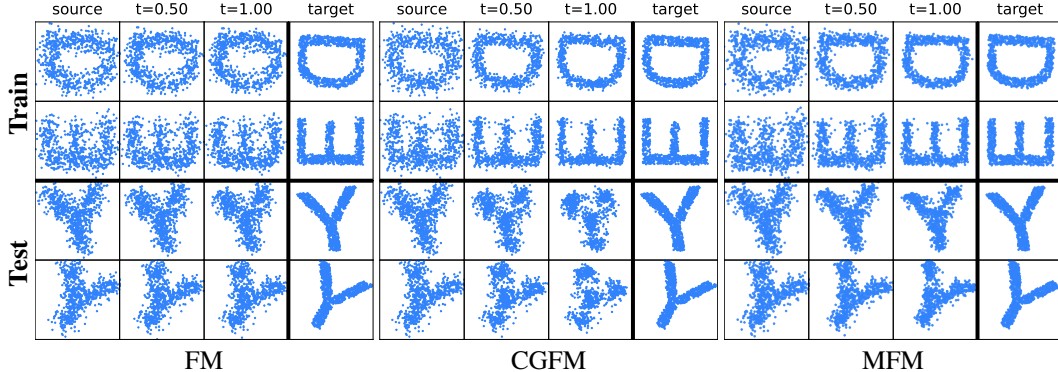

Figure 3: **Synthetic letters experiment visualizations.** Examples of model-generated samples from the source distribution ($t = 0$) to predicted target distribution ($t = 1$). See Fig. 6 Appendix F for further examples.

Table 1: **Results of the synthetic letters experiment for population prediction on seen train populations and unseen test populations.** We report the 1-Wasserstein ($\mathcal{W}_1$), 2-Wasserstein ($\mathcal{W}_2$), and the maximum-mean-discrepancy (MMD) distributional distances. We consider 4 settings for MFM with varying $k$. We use $^{w/}\mathcal{N}$ to denote models that use Gaussian source distributions sampled via $x_0 \sim \mathcal{N}(\mathbf{0}, \mathbf{1})$, but maintains the noisy letters source populations $p_0$ as input to the population embedding model $\varphi(p_0; \theta, k)$. Populations that comprise test sets (X and Y) are entirely unseen during training. We include an ablation on the effect of changing the number of training populations (Fig. 5) and results when using OT couplings between samples (Table 4) in Appendix F.

| | Train | | | Test (X's) | | | Test (Y's) | | |
|---|---|---|---|---|---|---|---|---|---|
| | $\mathcal{W}_1(\downarrow)$ | $\mathcal{W}_2(\downarrow)$ | MMD $_{(\times 10^{-3})}(\downarrow)$ | $\mathcal{W}_1(\downarrow)$ | $\mathcal{W}_2(\downarrow)$ | MMD $_{(\times 10^{-3})}(\downarrow)$ | $\mathcal{W}_1(\downarrow)$ | $\mathcal{W}_2(\downarrow)$ | MMD $_{(\times 10^{-3})}(\downarrow)$ |
| FM | $0.209 \pm 0.000$ | $0.277 \pm 0.000$ | $2.54 \pm 0.00$ | $0.234 \pm 0.000$ | $0.309 \pm 0.000$ | $2.45 \pm 0.00$ | $0.238 \pm 0.000$ | $0.316 \pm 0.000$ | $3.32 \pm 0.01$ |
| FM$^{w/}\mathcal{N}$ | $0.806 \pm 0.000$ | $0.960 \pm 0.000$ | $31.68 \pm 0.00$ | $0.764 \pm 0.000$ | $0.931 \pm 0.000$ | $25.04 \pm 0.00$ | $1.030 \pm 0.000$ | $1.228 \pm 0.000$ | $45.36 \pm 0.00$ |
| CGFM | $0.090 \pm 0.000$ | $0.113 \pm 0.000$ | $0.25 \pm 0.00$ | $0.334 \pm 0.000$ | $0.407 \pm 0.000$ | $5.55 \pm 0.00$ | $0.327 \pm 0.000$ | $0.405 \pm 0.000$ | $6.85 \pm 0.00$ |
| CGFM$^{w/}\mathcal{N}$ | $0.156 \pm 0.025$ | $0.201 \pm 0.027$ | $1.02 \pm 0.39$ | $0.849 \pm 0.004$ | $0.993 \pm 0.003$ | $35.08 \pm 0.75$ | $1.062 \pm 0.011$ | $1.229 \pm 0.010$ | $55.66 \pm 0.76$ |
| MFM$_{k=0}$ $^{w/}\mathcal{N}$ (ours) | $0.148 \pm 0.003$ | $0.195 \pm 0.010$ | $0.94 \pm 0.11$ | $0.347 \pm 0.011$ | $0.431 \pm 0.012$ | $6.47 \pm 0.44$ | $0.402 \pm 0.011$ | $0.485 \pm 0.010$ | $10.92 \pm 0.18$ |
| MFM$_{k=1}$ $^{w/}\mathcal{N}$ (ours) | $0.154 \pm 0.004$ | $0.208 \pm 0.010$ | $0.91 \pm 0.01$ | $0.349 \pm 0.023$ | $0.433 \pm 0.023$ | $6.53 \pm 0.52$ | $0.391 \pm 0.035$ | $0.477 \pm 0.041$ | $10.71 \pm 1.86$ |
| MFM$_{k=10}$ $^{w/}\mathcal{N}$ (ours) | $0.151 \pm 0.013$ | $0.197 \pm 0.015$ | $0.94 \pm 0.15$ | $0.343 \pm 0.020$ | $0.427 \pm 0.019$ | $6.38 \pm 0.67$ | $0.413 \pm 0.018$ | $0.502 \pm 0.024$ | $11.93 \pm 1.14$ |
| MFM$_{k=50}$ $^{w/}\mathcal{N}$ (ours) | $0.174 \pm 0.005$ | $0.232 \pm 0.006$ | $1.40 \pm 0.13$ | $0.363 \pm 0.018$ | $0.449 \pm 0.013$ | $7.46 \pm 0.44$ | $0.446 \pm 0.021$ | $0.536 \pm 0.028$ | $13.40 \pm 0.23$ |
| MFM$_{k=0}$ (ours) | $\mathbf{0.081 \pm 0.003}$ | $\mathbf{0.100 \pm 0.004}$ | $\mathbf{0.16 \pm 0.06}$ | $0.202 \pm 0.002$ | $0.249 \pm 0.003$ | $2.29 \pm 0.05$ | $0.218 \pm 0.001$ | $0.262 \pm 0.002$ | $3.79 \pm 0.11$ |
| MFM$_{k=1}$ (ours) | $0.082 \pm 0.001$ | $0.101 \pm 0.002$ | $\mathbf{0.16 \pm 0.01}$ | $0.205 \pm 0.008$ | $0.251 \pm 0.008$ | $2.38 \pm 0.22$ | $0.215 \pm 0.006$ | $0.258 \pm 0.007$ | $3.78 \pm 0.25$ |
| MFM$_{k=10}$ (ours) | $0.088 \pm 0.002$ | $0.109 \pm 0.003$ | $0.21 \pm 0.01$ | $\mathbf{0.201 \pm 0.006}$ | $\mathbf{0.248 \pm 0.006}$ | $2.20 \pm 0.15$ | $0.208 \pm 0.003$ | $0.252 \pm 0.002$ | $3.55 \pm 0.06$ |
| MFM$_{k=50}$ (ours) | $0.092 \pm 0.004$ | $0.116 \pm 0.004$ | $0.25 \pm 0.06$ | $0.206 \pm 0.008$ | $0.257 \pm 0.008$ | $\mathbf{2.18 \pm 0.25}$ | $\mathbf{0.204 \pm 0.005}$ | $\mathbf{0.249 \pm 0.006}$ | $\mathbf{3.14 \pm 0.18}$ |

diffusion process and learn the push-forward map from $p_0(x_0 \mid i)$ to $p_1(x_1 \mid i)$ for every index $i$. For this task, given the $i$-th population index we denote $p_0(x_0 \mid i)$ as the *source* population $p_1(x_1 \mid i)$ as the $i$-th *target* population.

To construct $p_1(x_1 \mid i)$, we discretize samples from a defined silhouette; e.g. an image of a character, where $i$ indexes the respective character. We use upper case letters as the silhouette and generate the corresponding samples $x_1 \sim p_1(x_1 \mid i)$ from the uniform distribution over the silhouette and run the diffusion process for samples $x_1$ to acquire $x_0$. We construct the *training data* using 10 random orientations of 24 letters. We construct the *test data* by using 10 random orientations of "X" and "Y". Test data populations of "X" and "Y" are entirely unseen during training.

We train FM, CGFM and 4 variants of MFM of varying $k$ for the GCN population embedding model $\varphi(p_0; \theta, k)$. When $k = 0$, $\varphi(p_t; \theta, k)$ becomes identical to the DeepSets model (Zaheer et al., 2017). We compare MFM to FM and CGFM. We repeat this experiment for models trained with source distributions sampled from a standard normal $x_0 \sim \mathcal{N}(\mathbf{0}, \mathbf{1})$. We label these models as FM$^{w/}\mathcal{N}$, CGFM$^{w/}\mathcal{N}$, and MFM$^{w/}\mathcal{N}$. Here, MFM$^{w/}\mathcal{N}$ still takes the original $p_0$ as input to the population embedding model $\varphi(p_0; \theta, k)$, while $v_t(\cdot; \omega)$ uses $x_0 \sim \mathcal{N}(\mathbf{0}, \mathbf{1})$ as the source distribution.

**MFM generalizes to populations from unseen letter silhouettes.** FM does not have access to conditional information; hence will only learn an aggregated lens of the distribution dynamics and will not be able to fit the training data, and consequently won't generalize to the test conditions. For the training data, the CGFM vector field model takes in the distribution index $i$ as a one-hot input condition. On the test set, since none of these indices is present, we input the normalized constant vector, which averages the learned embeddings of the indices. Because of this, CGFM will fit the training data, however, will not be able to generalize to the unseen condition in the test dataset. CGFM indicates when the model can fit the training data and demonstrates that the test data is substantially different and cannot be generated by the same model. For MFM, we expect to both fit the training data and generalize to unseen distributional conditions.

We report results for the synthetic experiment in Fig. 3 and Table 1. We observe that indeed FM struggles to adequately learn to sample from $p_1(x_1 \mid i)$ in the training set, while CGFM is able to

effectively sample from $p_1(x_1 \mid i)$ in the training set. As expected, CGFM fits the training data, but struggles to generalize beyond its set of training conditions. In contrast, we see that MFM is able to both fit the training data while also generalizing to the unseen test distributions. Interestingly, although MFM performs better for certain values of $k$ versus others, performance does not vary significantly for the range considered. In Fig. 7 (Appendix F.1) we plot the letter population embeddings.

## 5.2 EXPERIMENTS ON ORGANOID DRUG-SCREEN DATA

**Organoid drug-screen data.** For experiments on biological data, we use the organoid drug-screen dataset from Ramos Zapatero et al. (2023). This dataset is a single-cell mass-cytometry dataset collected over 10 patients. Somewhat unique to this dataset, unlike many prior perturbation-screen datasets which have a single control population, this dataset has matched controls to each experimental condition. Populations from each patient are treated with 11 different drug treatments of varying dose concentrations.[2] We use the term *replicate* to define control-treatment population pairs, $p_0(x_0 \mid c_i)$ and $p_1(x_1 \mid c_i)$, respectively (see Fig. 4-left).

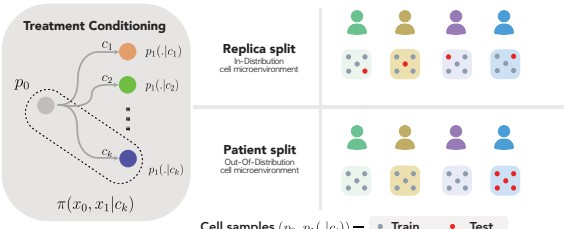

Figure 4: **Organoid drug-screen dataset overview.** (*Left*) a given replica consists of a control distribution $p_0$ and corresponding treatment response distribution $p_1$ for treatment condition $c_i$. (*Right*) train and test data splits for replicates (top) and patients (bottom). Experiments are conducted for 11 treatments, 10 patients, 3 culture conditions, and repeated (replicated) numerous times, resulting in a dataset of many control and treated population pairs.

After pre-processing, we acquire a total of 927 population pairs. In each patient, cell population are categorized into 3 cell *cultures*: (i) cancer associated Fibroblasts, (ii) patient-derived organoid cancer cells (PDO), and (iii) patient-derived organoid cancer cells co-cultured fibroblasts (PDOF). We report results for the individual cultures, i.e. Fibroblast, PDO, and PDOF cultures, in Appendix F.

**Pre-processing and data splits.** We filter each cell population to contain at least 1000 cells and use 43 bio-markers. We construct two data splits for the organoid drug-screen dataset (see Fig. 4-right). *Replicate split*; here we consider 2 settings, (i) leave-out replicates evenly across all patients for testing (*replica-1*, depicted in Fig. 4 see Table 5 for results on this split), and (ii) leaving out a batch of replicates from a single patient while including the remaining replicas from the same patient for training (*replica-2*, results in Table 2). The second setting is the *Patients split* (results in Table 3); here we leave-out replicates fully in one patients – in this setting, we are testing the ability of the model to generalize population prediction of treatment response for unseen patients. We do this for 3 different patients and report results across these independent splits. Further details regarding the organoid drug-screen dataset, data pre-processing, and data splits are provided in Appendix D.2.

Table 2: **Experimental results on the organoid drug-screen dataset** for population prediction of treatment response over *replicates*. We use $^{w/}$OT to denote models that incorporate OT couplings between source and target distributions. Note, the ICNN model operates in the OT regime. We use $^{w/}\mathcal{N}$ to denote models that use Gaussian source distributions sampled via $x_0 \sim \mathcal{N}(\mathbf{0}, \mathbf{1})$, but maintain the control populations $p_0$ as input to the population embedding model $\varphi(p_0; \theta, k)$. We denote the best non-OT method with **bold** and the best OT-based method with underline.

| | | $\mathcal{W}_1(\downarrow)$ | $\mathcal{W}_2(\downarrow)$ | MMD $_{(\times 10^{-3})}(\downarrow)$ | $r^2(\uparrow)$ |
|---|---|---|---|---|---|
| $d(p_0, p_1)$ | | 4.513 | 4.695 | 19.14 | 0.876 |
| $d(p_0 - \mu_0 + \tilde{\mu}_1, p_1)$ | | 6.222 | 6.346 | 74.90 | 0.876 |
| FM | | $4.340 \pm 0.078$ | $4.564 \pm 0.111$ | $13.00 \pm 0.67$ | $0.865 \pm 0.034$ |
| CGFM | | $4.443 \pm 0.033$ | $4.621 \pm 0.041$ | $17.00 \pm 1.03$ | $0.899 \pm 0.008$ |
| MFM$_{k=0}$ | (ours) | $4.209 \pm 0.007$ | $4.380 \pm 0.012$ | $12.34 \pm 0.50$ | $\mathbf{0.918 \pm 0.002}$ |
| MFM$_{k=10}$ | (ours) | $4.216 \pm 0.090$ | $4.395 \pm 0.098$ | $11.99 \pm 2.36$ | $0.917 \pm 0.005$ |
| MFM$_{k=50}$ | (ours) | $4.214 \pm 0.017$ | $4.396 \pm 0.020$ | $12.09 \pm 0.75$ | $0.916 \pm 0.002$ |
| MFM$_{k=100}$ | (ours) | $\mathbf{4.100 \pm 0.093}$ | $\mathbf{4.269 \pm 0.104}$ | $\mathbf{8.96 \pm 1.88}$ | $0.917 \pm 0.004$ |
| FM$^{w/}\mathcal{N}$ | | $7.114 \pm 0.100$ | $7.404 \pm 0.086$ | $64.97 \pm 3.79$ | $0.613 \pm 0.008$ |
| CGFM$^{w/}\mathcal{N}$ | | $7.135 \pm 0.045$ | $7.390 \pm 0.037$ | $79.78 \pm 4.67$ | $0.637 \pm 0.010$ |
| MFM$_{k=0}$ $^{w/}\mathcal{N}$ | (ours) | $4.177 \pm 0.042$ | $4.355 \pm 0.048$ | $10.53 \pm 0.59$ | $0.911 \pm 0.001$ |
| MFM$_{k=10}$ $^{w/}\mathcal{N}$ | (ours) | $4.156 \pm 0.065$ | $4.324 \pm 0.067$ | $9.58 \pm 1.63$ | $0.912 \pm 0.003$ |
| MFM$_{k=50}$ $^{w/}\mathcal{N}$ | (ours) | $4.153 \pm 0.069$ | $4.324 \pm 0.070$ | $9.63 \pm 1.45$ | $0.912 \pm 0.002$ |
| MFM$_{k=100}$ $^{w/}\mathcal{N}$ | (ours) | $4.166 \pm 0.001$ | $4.341 \pm 0.003$ | $9.52 \pm 0.33$ | $0.915 \pm 0.005$ |
| FM$^{w/}$OT | | $\underline{4.210 \pm 0.006}$ | $\underline{4.397 \pm 0.001}$ | $\underline{12.16 \pm 0.72}$ | $\underline{0.910 \pm 0.005}$ |
| CGFM$^{w/}$OT | | $4.356 \pm 0.027$ | $4.531 \pm 0.025$ | $15.82 \pm 0.19$ | $0.909 \pm 0.003$ |
| ICNN | | $4.488 \pm 0.035$ | $4.665 \pm 0.038$ | $17.60 \pm 0.55$ | $0.884 \pm 0.002$ |

**Baselines.** For the organoid drug-screen experiments, we also consider an ICNN model as a baseline as well as the OT variations of FM and CGFM. The ICNN is equivalent to the architecture used in CellOT (Bunne et al., 2023); a method for learning cell specific response to treatments. The ICNN (and likewise CellOT) counterparts our FM$^{w/}$OT model in that it does not take the population index $i$ as a condition. CGFM takes in the population identity index $i$ as a one-hot input condition. Unlike CGFM, MFM does not require knowledge of population identities. We additionally include two model-agnostic baselines: (i) the distributional distances between the target and source distributions

---

[2] We consider only the highest dosage and leave exploration of dose-dependent response to future work.

$d(p_0, p_1)$, and (ii) the distributional distances between the target and the source distributions shifted by the average mean of target distributions ($\tilde{\mu}_1$) of the training data $d(p_0 - \mu_0 + \tilde{\mu}_1, p_1)$.

**Predicting treatment response across replicates.** We show results for generalization across replicates for the ***replica-2*** split in Table 2. In Appendix F, we include an extended evaluation as well as report results for separate cell cultures in Table 8 and Table 9. We observe that MFM outperforms all baselines on left-out replicas. Although this is in general a simpler generalization problem compared to the left-out patient populations setting, this result suggests that there is arguably sufficient biological heterogeneity across replicas for MFM to learn meaningful embeddings.

**Predicting treatment response across patients.** We show results for generalization across patients in Table 3. We observe that MFM (w/o OT) outperforms all non-OT baseline methods as well as the ICNN (which learns OT couplings), while yielding competitive performance to the FM^w/OT and CGFM^w/OT. All methods yield generally equal degrees of variation across 3 patient splits, with comparable variation relative to the trivial baseline. See Appendix F.1 for an analysis of population embeddings.

Table 3: **Experimental results on the organoid drug-screen dataset** for population prediction of treatment response across left-out ***patient*** populations. We report the mean and standard deviations across metrics computed over 3 different patient splits. We denote the best non-OT method with **bold** and the best OT-based method with underline.

| | | $\mathcal{W}_1(\downarrow)$ | $\mathcal{W}_2(\downarrow)$ | MMD $_{(\times 10^{-3})}(\downarrow)$ | $r^2(\uparrow)$ |
|---|---|---|---|---|---|
| $d(p_0, p_1)$ | | $4.175 \pm 0.135$ | $4.303 \pm 0.174$ | $12.11 \pm 2.07$ | $0.902 \pm 0.006$ |
| $d(p_0 - \mu_0 + \tilde{\mu}_1, p_1)$ | | $6.158 \pm 0.239$ | $6.235 \pm 0.229$ | $77.74 \pm 10.72$ | $0.902 \pm 0.006$ |
| FM | | $4.171 \pm 0.107$ | $4.315 \pm 0.142$ | $10.95 \pm 1.98$ | $0.897 \pm 0.023$ |
| CGFM | | $4.189 \pm 0.088$ | $4.321 \pm 0.119$ | $11.57 \pm 0.96$ | $0.914 \pm 0.008$ |
| $\text{MFM}_{k=0}$ | (ours) | $4.135 \pm 0.094$ | $4.268 \pm 0.128$ | $10.18 \pm 1.28$ | $0.918 \pm 0.007$ |
| $\text{MFM}_{k=10}$ | (ours) | $4.112 \pm 0.086$ | $4.243 \pm 0.121$ | $9.90 \pm 0.99$ | $0.925 \pm 0.008$ |
| $\text{MFM}_{k=50}$ | (ours) | $\mathbf{4.087 \pm 0.122}$ | $\mathbf{4.218 \pm 0.160}$ | $\mathbf{9.26 \pm 1.56}$ | $0.926 \pm 0.007$ |
| $\text{MFM}_{k=100}$ | (ours) | $4.112 \pm 0.148$ | $4.244 \pm 0.186$ | $9.63 \pm 2.08$ | $\mathbf{0.931 \pm 0.002}$ |
| FM^w/OT | | $\underline{4.064 \pm 0.152}$ | $\underline{4.189 \pm 0.194}$ | $\underline{9.44 \pm 2.49}$ | $\underline{0.932 \pm 0.005}$ |
| CGFM^w/OT | | $4.087 \pm 0.129$ | $4.217 \pm 0.165$ | $9.83 \pm 2.03$ | $0.924 \pm 0.009$ |
| ICNN | | $4.157 \pm 0.168$ | $4.282 \pm 0.213$ | $11.18 \pm 2.51$ | $0.904 \pm 0.005$ |

Through the biological and synthetic experiments, we have shown that MFM is able to generalize to unseen distributions. The implication of our results suggest that MFM can learn population dynamics in unseen environments. In biological contexts, like the one we have shown in this work, this result indicates that we can improve learning of population dynamics, of treatment response or any arbitrary perturbation, in new/unseen patients. This works towards a model where it is possible to predict and design an individualized treatment regimen for each patient based on their individual characteristics and tumor microenvironment.

## 6 CONCLUSION

Our paper highlights the significance of modeling dynamics based on the entire distribution. While flow-based models offer a promising avenue for learning dynamics at the population level, they were previously restricted to a single initial population and predefined (known) conditions. In this paper, we introduce *Meta Flow Matching* (MFM) as a practical solution to address these limitations. By integrating along vector fields of the Wasserstein manifold, MFM allows for a more comprehensive model of dynamical systems with interacting particles. Crucially, MFM leverages graph neural networks to embed the initial population, enabling the model to generalize over various initial distributions. MFM opens up new possibilities for understanding complex phenomena that emerge from interacting systems in biological and physical systems. In practice, we demonstrate that MFM learns meaningful embeddings of single-cell populations along with the developmental model of these populations. Moreover, our empirical study demonstrates the possibility of modeling patient-specific response to treatments via the meta-learning. We discuss broader impacts of this work in Appendix C.

**Limitations & Future work.** In this work, we focused on exploring the amortization of learning over the space of distributions using flow matching. We argue this is a more natural model for many biological systems. However, there are many other aspects of modeling biological systems that we did not consider. In particular we did not consider extensions to the manifold setting (Huguet et al., 2022; 2023), unbalanced optimal transport (Benamou, 2003; Yang and Uhler, 2019; Chizat et al., 2018), aligned (Somnath et al., 2023; Liu et al., 2023), or stochastic settings (Bunne et al., 2023; Koshizuka and Sato, 2022) in this work. We observed that OT improves performance of FM and CGFM in the biological settings which we considered. As such, we view investigating the use of OT in MFM as a natural future direction. Since the focus of this work was to investigate the effect of conditioning flow models on initial distributions, we leave this exploration for future work. We also note that our framework can be extended beyond flow matching. One of the central innovations in this work is conditioning vector fields on representations of entire distributions. Hence, this can be easily extended to other training regimes. Furthermore, we proposed the joint training of parameters $\omega$ and $\theta$ for the vector field $v_t(\cdot; \omega)$ and population embedding model $\varphi(\cdot; \theta)$ using the loss defined in Eq. (17). Although this is a novel approach to $v_t(\cdot|\varphi(\cdot; \theta); \omega)$ training, there remains room to explore the pre-training and/or use of different losses for the population embedding model.

## REPRODUCIBILITY STATEMENT

To ensure reproducibility of our findings and results we submitted our source code as supplementary materials. In addition, we describe the mathematical details of our method throughout Section 3 and provide pseudo-code to reproduce training and inference in Algorithm 1 and Algorithm 2. We provide details regarding parameterization of models, optimization, and implementation in Appendix D.4 and Appendix E. For our datasets, we include details regarding the synthetic letters dataset and how to construct in it Section 5.1 and Appendix D.1, and details for data download and data processing for the Organoid drug-screen data in Section 5.2 and Appendix D.2. Information regarding the datasets that we consider in this work can also be found in the source code. For our theoretical contributions, we state our assumptions throughout the text in Section 2 and Section 3 and provide detailed proofs for Proposition 1 (in the main text) and Theorem 1 (in Appendix B).

## ACKNOWLEDGMENTS

The authors acknowledge funding from UNIQUE, CIFAR, NSERC, Intel, and Samsung. The research was enabled in part by by the Province of Ontario and companies sponsoring the Vector Institute (http://vectorinstitute.ai/partners/), the computational resources provided by the Digital Research Alliance of Canada (https://alliancecan.ca), Mila (https://mila.quebec), and NVIDIA. KN is supported by IVADO.

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

## A DEFINITIONS

All the densities we consider are the densities of the probability measures from the following set

$$\mathcal{P}_2(\mathcal{X}) := \left\{ \mu \in \mathcal{P}(\mathcal{X}) : \int_{\mathcal{X}} d\mu(x) \, d(x, \bar{x})^2 < +\infty \quad \text{for some} \quad \bar{x} \in \mathcal{X} \right\}, \tag{20}$$

where $\mathcal{P}(\mathcal{X})$ is the family of all Borel probability measures on $\mathcal{X}$.

Thus, one can introduce 2-Wasserstein distance between measures in $\mathcal{P}_2(\mathcal{X})$, i.e.

$$W_2^2(\mu_0, \mu_1) := \min\left\{ \int_0^1 dt \, \|v_t\|^2_{L^2(p_t;\mathcal{X})} : \frac{\partial p_t}{\partial t} + \langle \nabla, p_t v_t \rangle = 0 \right\}, \tag{21}$$

where we used the Benamou-Brenier formulation of optimal transport (Benamou, 2003). That is, the possible changes of the density are restricted by the continuity equation

$$\frac{\partial p_t(x)}{\partial t} + \langle \nabla_x, p_t(x) v_t(x) \rangle = 0 \,, \tag{22}$$

where the define the divergence as

$$\langle \nabla_x, v(x) \rangle = \sum_{i=1}^{d} \frac{\partial v_i(x)}{\partial x_i} \,. \tag{23}$$

Note that $W_2^2(\cdot, \cdot)$ define a metric on $\mathcal{P}_2(\mathcal{X})$, which allows to introduce absolutely-continuous curves and the following tangent space

$$\mathrm{Tan}_\mu \mathcal{P}_2(\mathcal{X}) := \overline{\{\nabla s : s \in C^\infty(\mathcal{X})\}}^{L^2(\mu;\mathcal{X})} \quad \forall \mu \in \mathcal{P}_2(\mathcal{X}) \,. \tag{24}$$

These developments lead to the celebrated Riemannian interpretation of the 2-Wasserstein distance $W_2^2(\cdot, \cdot)$ on $\mathcal{P}_2(\mathcal{X})$ proposed by (Otto, 2001) and rigorously studied and presented in (Ambrosio et al., 2008).

## B PROOF OF THEOREM 1

**Theorem 1.** *Consider a dataset of populations $\mathcal{D} = \{(\pi(x_0, x_1 \mid i))\}_i$ generated from some unknown conditional model $\pi(x_0, x_1 \mid c)p(c)$. Then the following objective*

$$\mathcal{L}(\omega, \theta) = \mathbb{E}_{p(c)} \int_0^1 dt \, \mathbb{E}_{p_t(x_t \mid c)} \|v_t^*(x_t \mid c) - v_t(x_t \mid \varphi(p_0, \theta), \omega)\|^2 \tag{16}$$

*is equivalent to the Meta Flow Matching objective*

$$\mathcal{L}_{MFM}(\omega, \theta) = \mathbb{E}_{i \sim \mathcal{D}} \mathbb{E}_{\pi(x_0, x_1 \mid i)} \int_0^1 dt \, \left\| \frac{\partial}{\partial t} f_t(x_0, x_1) - v_t(f_t(x_0, x_1) \mid \varphi(p_0, \theta); \omega) \right\|^2 \tag{17}$$

*up to an additive constant.*

*Proof.* The loss function

$$\mathcal{L}(\omega, \theta) = \mathbb{E}_{p(c)} \int_0^1 dt \, \mathbb{E}_{p_t(x_t \mid c)} \|v_t^*(x_t \mid c) - v_t(x_t \mid \varphi(p_t, \theta); \omega)\|^2 \tag{25}$$

$$= -2\mathbb{E}_{p(c)} \int dt dx \, \langle p_t(x \mid c) v_t^*(x \mid c), v_t(x \mid \varphi(p_t, \theta); \omega) \rangle + \tag{26}$$

$$+ \mathbb{E}_{p(c)} \int_0^1 dt \, \mathbb{E}_{p_t(x_t \mid c)} \|v_t(x_t \mid \varphi(p_t, \theta), \omega)\|^2 + \tag{27}$$

$$+ \mathbb{E}_{p(c)} \int_0^1 dt \, \mathbb{E}_{p_t(x_t \mid c)} \|v_t^*(x_t \mid c)\|^2 \,. \tag{28}$$

The last term does not depend on $\theta$, the second term we can estimate, for the first term, we use the formula for the (from Eq. (8))

$$p_t(\xi \mid c) v_t^*(\xi \mid c) = \mathbb{E}_{\pi(x_0, x_1)} \delta(f_t(x_0, x_1) - \xi) \frac{\partial f_t(x_0, x_1)}{\partial t} \,. \tag{29}$$

Thus, the loss is equivalent (up to a constant) to

$$\mathcal{L}(\omega, \theta) = -2\mathbb{E}_{p(c)}\mathbb{E}_{\pi(x_0, x_1 \,|\, c)} \int dt \left\langle \frac{\partial f_t(x_0, x_1)}{\partial t}, v_t(f_t(x_0, x_1) \,|\, \varphi(p_t, \theta); \omega) \right\rangle + \tag{30}$$

$$+ \mathbb{E}_{p(c)}\mathbb{E}_{\pi(x_0, x_1 \,|\, c)} \int_0^1 dt \, \|v_t(f_t(x_0, x_1) \,|\, \varphi(p_t, \theta), \omega)\|^2 \pm \tag{31}$$

$$\pm \mathbb{E}_{p(c)}\mathbb{E}_{\pi(x_0, x_1 \,|\, c)} \int_0^1 dt \left\| \frac{\partial f_t(x_0, x_1)}{\partial t} \right\|^2 \tag{32}$$

$$= \mathbb{E}_{c \sim p(c)}\mathbb{E}_{\pi(x_0, x_1 \,|\, c)} \int_0^1 dt \left\| \frac{\partial}{\partial t} f_t(x_0, x_1) - v_t(f_t(x_0, x_1) \,|\, \varphi(p_t, \theta); \omega) \right\|^2 . \tag{33}$$

Note that in the final expression we do not need access to the probabilistic model of $p(c)$ if the joints $\pi(x_0, x_1 \,|\, c)$ are already sampled in the data $\mathcal{D}$. Thus, we have

$$\mathcal{L}(\omega, \theta) = \mathbb{E}_{c \sim p(c)}\mathbb{E}_{\pi(x_0, x_1 \,|\, c)} \int_0^1 dt \left\| \frac{\partial}{\partial t} f_t(x_0, x_1) - v_t(f_t(x_0, x_1) \,|\, \varphi(p_t, \theta); \omega) \right\|^2 \tag{34}$$

$$= \mathbb{E}_{i \sim \mathcal{D}}\mathbb{E}_{\pi(x_0, x_1 \,|\, i)} \int_0^1 dt \left\| \frac{\partial}{\partial t} f_t(x_0, x_1) - v_t(f_t(x_0, x_1) \,|\, \varphi(p_t, \theta); \omega) \right\|^2 \tag{35}$$

$$= \mathcal{L}_{\text{MFM}}(\omega, \theta) . \tag{36}$$

$\square$

## C  BROADER IMPACTS

This paper is primarily a theoretical and methodological contribution with little societal impact. MFM can be used to better model dynamical systems of interacting particles and in particular cellular systems. Better modeling of cellular systems can potentially be used for the development of malicious biological agents. However, we do not see this as a significant risk at this time.

## D  EXPERIMENTAL DETAILS

### D.1  SYNTHETIC LETTERS DATA

The synthetic letters dataset (used for Fig. 3 and Table 1) contains 240 train populations a 10 test populations. Each population contains roughly between 750 and 2700 samples in this dataset.

### D.2  ORGANOID DRUG-SCREEN DATA

The *organoid drug-screen* dataset consists of patient derived organoids (PDOs) from 10 different patients (Ramos Zapatero et al., 2023).[3] For each patient, experiments are replicated and performed over varrying conditions. This results in a significant amount of coupled populations pairs $(p_0(x_0|c_i), p_1(x_1|c_i))$, where we use $c_i$ to denote the treatment condition corresponding to respective population pair. After pre-processing, we acquire a total of 927 replicates (or coupled population pairs). In the *replica-1 split*, we use 713 populations for training, 111 left-out population for validation, and 103 left-out populations for testing. In the *replica-2* split, we use 861 populations for training, 33 left-out populations for validation, and 33 left-out populations for testing. We use the replica splits to set and select reasonable hyperparameters for MFM and the baseline models.

For the patients split, we consider 3 different patient splits where we independently leave out all populations from either patient 21, patient 27, and patients 75. Organoid PDO-21 was found to present an interesting chemoprotective response (High CAF Protection) to cancer treatments, and hence was selected by Ramos Zapatero et al. (2023) for further study and data collection. For the same reason we select PDO-21 as the unseen test patient for the *patients split* experiments. We additionally consider populations from PDO-75 as a third left-out patient split. In the patients splits, we only split data into train and test sets, resulting in 839 population pairs for training and 88 population pairs for test evaluation for PDO-21 and PDO-27 splits. For PDO-75, the train split contain 830 train population pairs and 97 test population pairs.

---

[3]PDOs are cultures of cell populations which are derived from patient cells.

### D.3 METRICS

We use four metrics to evaluate the quality of a generated empirical distribution $\hat{p}$ as compared to an empirical ground truth distribution $p$. In this section we detail the calculation and interpretation of each metric. For letters, we evaluate over all samples, while for the organoid drug-screen dataset we evaluate over 5000 cells. If a target population contains less than 5000 cells, we evaluate using sample size corresponding to the size of the entire target population.

**1-Wasserstein ($\mathcal{W}_1$) and 2-Wasserstein ($\mathcal{W}_2$).** We evaluate the Wasserstein distance for $\alpha = 1$ and $\alpha = 2$. Specifically for $\{\hat{x}_i\}_{i=0}^n = \hat{p}$ and $\{x_i\}_{i=0}^n = p$ we calculate the $\alpha$-Wasserstein distance with respect to the Euclidean distance as

$$\mathcal{W}_\alpha = \left( \inf_{\pi \in \Pi(\hat{p}, p)} \int \|x - y\|_2^\alpha d\pi(x, y) \right)^{1/\alpha} \tag{37}$$

where $\Pi(\hat{p}, p)$ denotes the set of all joint probability measures with marginals $\hat{p}, p$.

**Maximum mean discrepancy (MMD).** We evaluate an estimate for MMD between empirical distributions $\{\hat{x}_i\}_{i=0}^n = \hat{p}$ and $\{x_i\}_{i=0}^n = p$ using the radial basis function (RBF) kernel $k(\cdot, \cdot; \gamma)$ as:

$$\text{MMD}(p, \hat{p}) = \frac{1}{n^2} \sum_i^n \sum_j^n k(\hat{x}_i, \hat{x}_j; \gamma) + \frac{1}{n^2} \sum_i^n \sum_j^n k(x_i, x_j; \gamma) - \frac{2}{n^2} \sum_i^n \sum_j^n k(\hat{x}_i, x_j; \gamma). \tag{38}$$

We estimate MMD for $\gamma = 2, 1, 0.5, 0.1, 0.01, 0.005$ and report the average.

**Coefficient of determination ($r^2$).** We adopt the $r^2$ metric, overall average correlation coefficient, from Bunne et al. (2023) used to compare predictions and observations. This metric is computed as follows. First, given two arrays $X \in \mathbb{R}^{n \times d}$ and $Y \in \mathbb{R}^{n \times d}$, where $n$ is the number of samples and $d$ is the number of features, we can compute the correlation as

$$\text{corr}(X, Y) = \frac{\sum_{i=1}^n (X_i - \mu_X)^{\mathrm{T}} (Y_i - \mu_Y)}{\sqrt{\sum_{i=1}^n (X_i - \mu_X)^{\mathrm{T}} (X_i - \mu_X)} \sqrt{\sum_{i=1}^n (Y_i - \mu_Y)^{\mathrm{T}} (Y_i - \mu_Y)}}, \tag{39}$$

where $\mu_X$ and $\mu_Y$ are the empirical means of $X$ and $Y$. We compute correlation across all points for $X$ (predicted samples) and $Y$ (target samples) individually

$$\Sigma_{ij}^X = \text{corr}(X_i, X_j), \quad \Sigma_{ij}^Y = \text{corr}(Y_i, Y_j). \tag{40}$$

Then, we consider the two arrays

$$S^X = \Sigma_{ij}^X \forall i < j, \quad S^Y = \Sigma_{ij}^Y \forall i < j. \tag{41}$$

The overall average correlation coefficient is reported as

$$r^2 = \text{corr}(S^X, S^Y). \tag{42}$$

We compute a $r^2$ for every population in the evaluation sets and report the average $r^2$ over this set. We take this evaluation directly from Bunne et al. (2023), which is a typical evaluation of single-cell prediction methods.

### D.4 MODEL ARCHITECTURES AND HYPERPARAMETERS

**ICNN** The ICNN baseline was constructed with two networks ICNN network $f(x)$ and $g(x)$, with non-negative leaky ReLU activation layers. $f(x)$ is used to minimize the transport distance and $g(x)$ is used to transport from source to target. It has four hidden units with width of 64, and a latent dimension of 50. Both networks uses Adam optimizer ($lr = 10^{-4}$, $\beta_1 = 0.5$, $\beta_2 = 0.9$). $g(x)$ is trained with an inner iteration of 10 for every iteration $f(x)$ is trained.

**Vector Field Models** All vector field models $v_t$ are parameterized with linear layers of 512 hidden units and SELU activation functions. For the synthetic experiments, we use 4 hidden layers. For the biological experiments, we use 7 hidden layers and skip connections across every layer. We found that this setup worked well for the biological experiments. The FM vector field model additionally takes a conditional input for the one-hot treatment encoding. CGFM takes the conditional input for the one-hot treatment conditions as well as a one-hot encoding for the population index condition $i$. The MFM vector field model takes population embedding conditions, that is output from the GCN, as input, as well as the treatment one-hot encoding. All vector field models use temporal embeddings for time and positional embeddings for the input samples. We did not sweep the size of this embeddings space and found that a temporal embedding and positional embeddings sizes of 128 worked sufficiently well.

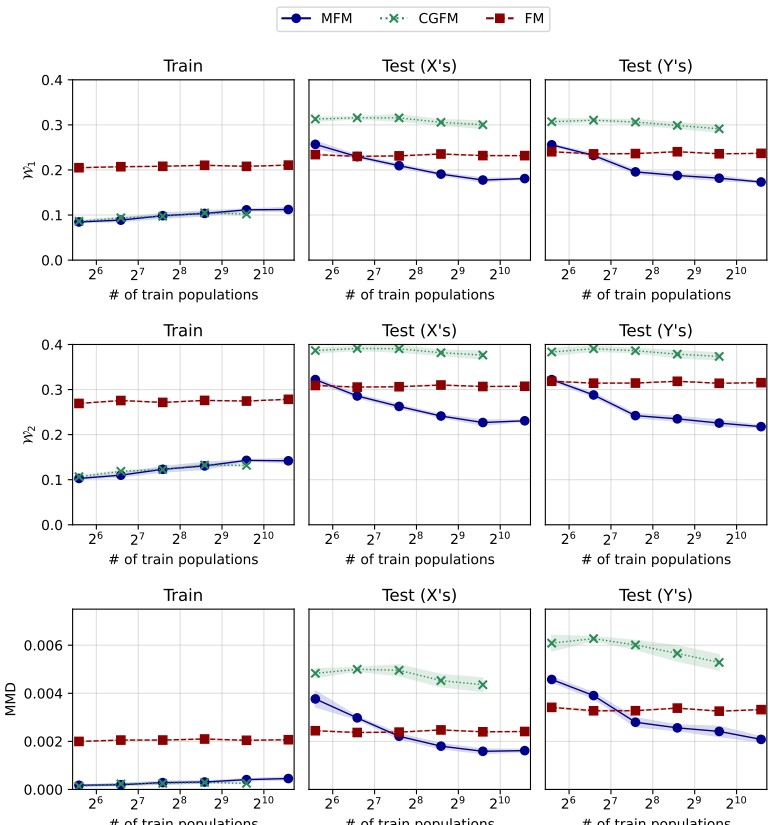

Figure 5: **Synthetic letters ablation over number of training populations.** Here we fix the test sets (X's and Y's) with 10 random rotations (same across each experiment). We then ablate the number of populations used for training FM (red), CGFM (green), and MFM (blue), by changing the number of random rotations/orientations used for each letter silhouette. We observe that for MFM the distributional errors on the test sets consistently decrease as we increase the number of training populations. In contrast, since FM and CGFM cannot generalize across novel populations, this increase in training populations does not lead to an overall improved performance on the test populations. For a large number of training populations, CGFM exhibits exhaustive memory requirements since it requires one-hot encodings as input conditions to denote the population index.

**Graph Neural Network** We considered a GCN model that consists of a $k$-nearest neighbour graph edge pooling layer and graph convolution layers with 512 hidden units. For the synthetic experiment we found that 3 GCN layers to work well, while for the biological experiments we found 2 GCN layers to perform well. The final GCN model layer outputs an embedding representation $e \in \mathbb{R}^d$. For the Synthetic experiment, we found that $d = 64$ performed well, and $d = 128$ performed well for the biological experiments. We normalize and project embeddings onto a hyper-sphere, and find that this normalization helps improve training. Additionally, the GCN takes a one-hot cell-type encoding (encoding for Fibroblast cells or PDO cells) for the control populations $p_0$. This may be beneficial for PDOF populations where both Fibroblast cells and PDO cells are present. However, it is important to note that labeling which cells are Fibroblasts versus PDOs withing the PDOF cultures is difficult and noisy in itself, hence such a cell-type condition may yield no additive information/performance gain.

**Optimization** We use the Adam optimizer with a learning rate of $0.0001$ for all Flow-matching models (FM, CGFM, MFM). We also used the Adam optimizer with a learning rate of $0.0001$ for the GCN model. To train the MFM (FM+GCN) models, we alternate between updating the vector field model parameters $\omega$ and the GCN model parameters $\theta$. We alternate between updating the respective model parameters every gradient step. For the synthethic experiment, FM and CGFM model were trained for 12000 epochs, while MFM models were trained for 24000 epochs, with a population batch size of 10 and a sample batch size of 700. Due to the alternating optimization, the MFM vector field model receives half as many updates compared to its counterparts (FM and CGFM). Therefore, training for the double the epochs is necessary for fair comparison.

**Influence of GCN parameter $k$ on performance.** We observe that no clear single selection for $k$ yields the best performance across all tasks on the single-cell experiments. For the *replicates* split, $k = 0$ on average performs better than $k > 0$, whereas on the *patients* split, the opposite is true, $k = 100$ performs best on average. This is possibly since the *patients* split forms a more difficult generalization problem (more diversity between training populations and test populations), and hence it is more difficult to over-fit during training with higher $k$.

The hyperparameters stated in this section were selected from brief and small grid search sweeps. We did not conduct any thorough hyperparameter optimization.

## E    IMPLEMENTATION DETAILS

We implement all our experiments using PyTorch and PyTorch Geometric. All experiments were conducted on a HPC cluster primarily on NVIDIA Tesla T4 16GB and A40 48GB GPUs. Each individual seed experiment run required only 1 GPU. Depending on the model (FM vs CGFM vs MFM) Each synthetic experiment ran between 6-24 hours and each real-data experiment ran between 12-36 hours. All experiments took approximately 500 GPU hours in aggregate.

# F EXTENDED RESULTS

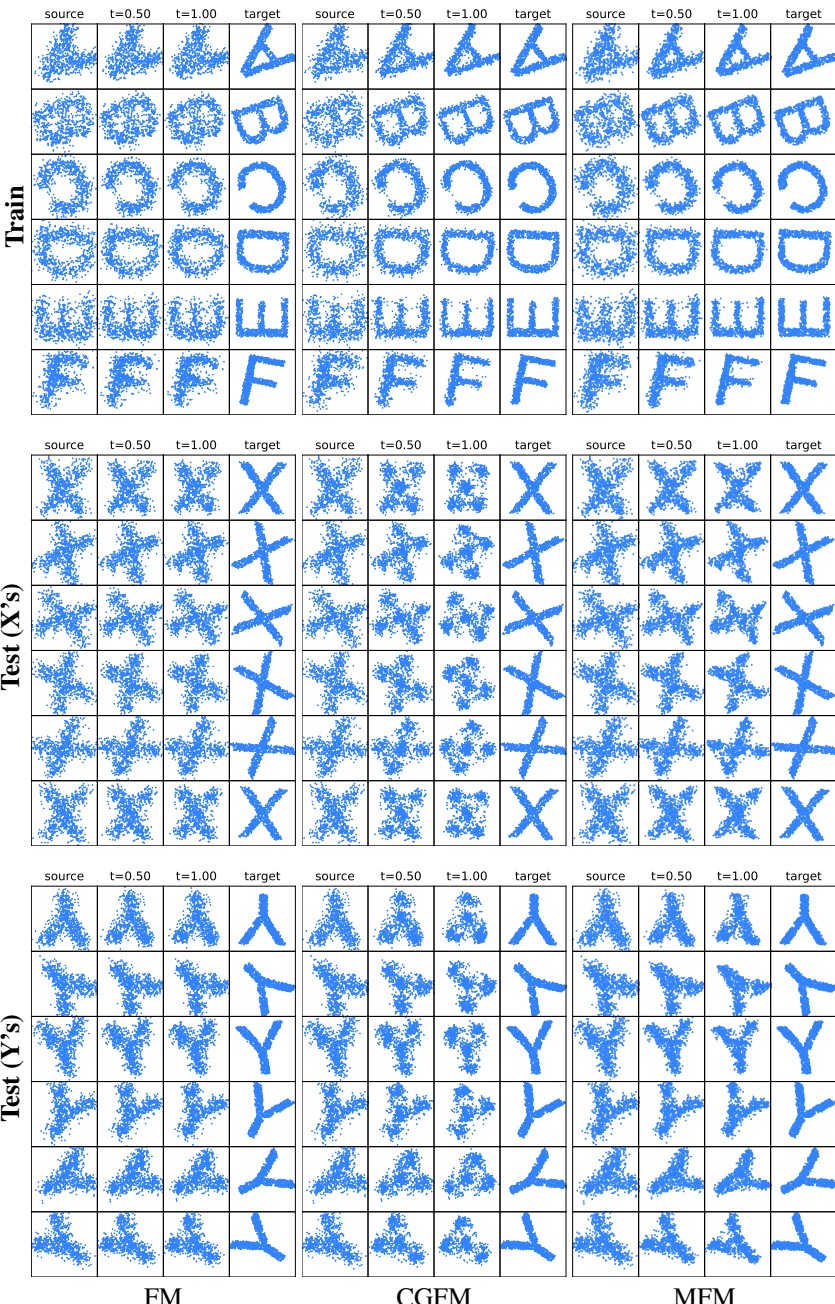

Figure 6: Model-generated samples for synthetic letters from the source ($t = 0$) to target ($t = 1$) distributions.

Table 4: **Comparison of models trained using independent coupling versus OT couplings on the synthetic letters setting.** We report the 1-Wasserstein ($\mathcal{W}_1$), 2-Wasserstein ($\mathcal{W}_2$), and the maximum-mean-discrepancy (MMD) distributional distances. We use $^{w/}$OT to denote models that incorporate OT couplings between source and target distributions (Tong et al., 2024a; Pooladian et al., 2023). We report results for MFM ($k = 50$). Here we report results for models trained with a population batch size of 1 and sample batch size of 700. We observe using OT couplings in this setting does not provide a significant change in performance.

| | Train | | | Test (X's) | | | Test (Y's) | | |
|---|---|---|---|---|---|---|---|---|---|
| | $\mathcal{W}_1(\downarrow)$ | $\mathcal{W}_2(\downarrow)$ | MMD $_{(\times 10^{-3})}(\downarrow)$ | $\mathcal{W}_1(\downarrow)$ | $\mathcal{W}_2(\downarrow)$ | MMD $_{(\times 10^{-3})}(\downarrow)$ | $\mathcal{W}_1(\downarrow)$ | $\mathcal{W}_2(\downarrow)$ | MMD $_{(\times 10^{-3})}(\downarrow)$ |
| FM | $0.210 \pm 0.000$ | $0.279 \pm 0.000$ | $2.548 \pm 0.000$ | $0.231 \pm 0.000$ | $0.307 \pm 0.000$ | $2.422 \pm 0.000$ | $0.236 \pm 0.000$ | $0.315 \pm 0.000$ | $3.305 \pm 0.000$ |
| FM$^{w/}$OT | $0.210 \pm 0.001$ | $0.277 \pm 0.001$ | $2.555 \pm 0.034$ | $0.234 \pm 0.001$ | $0.309 \pm 0.001$ | $2.458 \pm 0.030$ | $0.239 \pm 0.001$ | $0.316 \pm 0.001$ | $3.368 \pm 0.057$ |
| CGFM | $\mathbf{0.093 \pm 0.000}$ | $\mathbf{0.118 \pm 0.000}$ | $0.229 \pm 0.000$ | $0.321 \pm 0.000$ | $0.395 \pm 0.000$ | $5.125 \pm 0.000$ | $0.309 \pm 0.000$ | $0.388 \pm 0.000$ | $6.184 \pm 0.000$ |
| CGFM$^{w/}$OT | $\mathbf{0.093 \pm 0.000}$ | $0.119 \pm 0.000$ | $\mathbf{0.215 \pm 0.036}$ | $0.314 \pm 0.002$ | $0.390 \pm 0.002$ | $4.871 \pm 0.040$ | $0.307 \pm 0.001$ | $0.388 \pm 0.002$ | $6.021 \pm 0.084$ |
| MFM | $0.100 \pm 0.002$ | $0.127 \pm 0.002$ | $0.322 \pm 0.032$ | $\mathbf{0.189 \pm 0.003}$ | $\mathbf{0.238 \pm 0.005}$ | $1.742 \pm 0.086$ | $\mathbf{0.190 \pm 0.002}$ | $\mathbf{0.237 \pm 0.003}$ | $\mathbf{2.473 \pm 0.120}$ |
| MFM$^{w/}$OT | $0.103 \pm 0.007$ | $0.133 \pm 0.010$ | $0.450 \pm 0.234$ | $\mathbf{0.189 \pm 0.006}$ | $0.239 \pm 0.007$ | $\mathbf{1.706 \pm 0.139}$ | $0.193 \pm 0.006$ | $0.240 \pm 0.006$ | $2.522 \pm 0.220$ |

Table 5: **Experimental results on the organoid drug-screen dataset** for population prediction of treatment response across *replica-1* populations. Results shown in this table are for **dim = 43**. We use $^{w/}\mathcal{N}$ to denote models that use Gaussian source distributions sampled via $x_0 \sim \mathcal{N}(\mathbf{0}, \mathbf{1})$, but maintain the control populations $p_0$ as input to the population embedding model $\varphi(p_0; \theta, k)$. We further report $d(p_0 - \mu_0 + \mu_1, p_1)$, a baseline that uses the means from each true treated populations $p_1$.

| | Train | | | | Test | | | |
|---|---|---|---|---|---|---|---|---|
| | $\mathcal{W}_1(\downarrow)$ | $\mathcal{W}_2(\downarrow)$ | MMD $_{(\times 10^{-3})}(\downarrow)$ | $r^2(\uparrow)$ | $\mathcal{W}_1(\downarrow)$ | $\mathcal{W}_2(\downarrow)$ | MMD $_{(\times 10^{-3})}(\downarrow)$ | $r^2(\uparrow)$ |
| $d(p_0, p_1)$ | 4.182 | 4.294 | 10.84 | 0.925 | 4.100 | 4.222 | 10.21 | 0.931 |
| $d(p_0 - \mu_0 + \bar\mu_1, p_1)$ | 6.009 | 6.074 | 68.70 | 0.925 | 5.985 | 6.058 | 68.00 | 0.931 |
| $d(p_0 - \mu_0 + \mu_1, p_1)$ | 3.913 | 3.992 | 1.97 | 0.925 | – | – | – | – |
| FM | $3.925 \pm 0.019$ | $4.041 \pm 0.023$ | $3.76 \pm 0.26$ | $0.952 \pm 0.007$ | $3.961 \pm 0.036$ | $4.089 \pm 0.042$ | $5.90 \pm 0.25$ | $0.941 \pm 0.010$ |
| CGFM | $\mathbf{3.864 \pm 0.064}$ | $3.975 \pm 0.069$ | $3.16 \pm 0.89$ | $0.964 \pm 0.006$ | $4.087 \pm 0.063$ | $4.211 \pm 0.066$ | $8.84 \pm 0.75$ | $0.938 \pm 0.006$ |
| MFM ($k = 0$) | $3.874 \pm 0.015$ | $\mathbf{3.973 \pm 0.020}$ | $3.37 \pm 0.14$ | $\mathbf{0.967 \pm 0.003}$ | $\mathbf{3.880 \pm 0.009}$ | $\mathbf{3.990 \pm 0.011}$ | $4.68 \pm 0.16$ | $0.955 \pm 0.002$ |
| MFM ($k = 10$) | $3.896 \pm 0.021$ | $4.000 \pm 0.021$ | $3.82 \pm 0.12$ | $0.964 \pm 0.001$ | $3.899 \pm 0.013$ | $4.012 \pm 0.011$ | $5.13 \pm 0.48$ | $0.955 \pm 0.001$ |
| MFM ($k = 50$) | $3.888 \pm 0.038$ | $3.991 \pm 0.030$ | $3.59 \pm 0.41$ | $0.963 \pm 0.001$ | $3.900 \pm 0.038$ | $4.013 \pm 0.034$ | $5.06 \pm 0.22$ | $0.954 \pm 0.003$ |
| MFM ($k = 100$) | $3.906 \pm 0.010$ | $4.008 \pm 0.005$ | $4.05 \pm 0.38$ | $0.964 \pm 0.002$ | $3.898 \pm 0.008$ | $4.009 \pm 0.009$ | $5.19 \pm 0.05$ | $\mathbf{0.957 \pm 0.000}$ |
| FM$^{w/}\mathcal{N}$ | $6.908 \pm 0.037$ | $7.181 \pm 0.033$ | $57.70 \pm 0.75$ | $0.639 \pm 0.005$ | $6.972 \pm 0.022$ | $7.244 \pm 0.022$ | $60.39 \pm 0.98$ | $0.642 \pm 0.007$ |
| CGFM$^{w/}\mathcal{N}$ | $4.187 \pm 0.008$ | $4.340 \pm 0.009$ | $8.69 \pm 0.50$ | $0.936 \pm 0.002$ | $6.852 \pm 0.045$ | $7.114 \pm 0.044$ | $71.24 \pm 3.71$ | $0.666 \pm 0.016$ |
| MFM$^{w/}\mathcal{N}$ ($k = 0$) | $3.940 \pm 0.022$ | $4.047 \pm 0.023$ | $3.91 \pm 0.18$ | $0.959 \pm 0.006$ | $3.896 \pm 0.026$ | $4.002 \pm 0.030$ | $\mathbf{4.35 \pm 0.18}$ | $0.950 \pm 0.005$ |
| MFM$^{w/}\mathcal{N}$ ($k = 10$) | $3.976 \pm 0.044$ | $4.086 \pm 0.049$ | $4.52 \pm 0.42$ | $0.961 \pm 0.002$ | $3.943 \pm 0.032$ | $4.051 \pm 0.034$ | $5.28 \pm 0.25$ | $0.952 \pm 0.001$ |
| MFM$^{w/}\mathcal{N}$ ($k = 50$) | $3.968 \pm 0.013$ | $4.075 \pm 0.014$ | $4.36 \pm 0.44$ | $0.961 \pm 0.002$ | $3.934 \pm 0.007$ | $4.041 \pm 0.008$ | $4.99 \pm 0.35$ | $0.954 \pm 0.000$ |
| MFM$^{w/}\mathcal{N}$ ($k = 100$) | $3.937 \pm 0.014$ | $4.040 \pm 0.015$ | $3.94 \pm 0.00$ | $0.963 \pm 0.001$ | $3.908 \pm 0.030$ | $4.011 \pm 0.033$ | $4.68 \pm 0.52$ | $0.953 \pm 0.002$ |
| ICNN | $4.286 \pm 0.018$ | $4.313 \pm 0.112$ | $38.60 \pm 0.21$ | $0.897 \pm 0.031$ | $4.194 \pm 0.110$ | $4.313 \pm 0.112$ | $37.90 \pm 2.84$ | $0.897 \pm 0.008$ |

Table 6: **Experimental results on the organoid drug-screen dataset** for population prediction of treatment response across *replica-2* populations. Results shown in this table are for **dim = 43**.

| | Train | | | | Test | | | |
|---|---|---|---|---|---|---|---|---|
| | $\mathcal{W}_1(\downarrow)$ | $\mathcal{W}_2(\downarrow)$ | MMD $_{(\times 10^{-3})}(\downarrow)$ | $r^2(\uparrow)$ | $\mathcal{W}_1(\downarrow)$ | $\mathcal{W}_2(\downarrow)$ | MMD $_{(\times 10^{-3})}(\downarrow)$ | $r^2(\uparrow)$ |
| $d(p_0, p_1)$ | 4.096 | 4.213 | 8.66 | 0.923 | 4.513 | 4.695 | 19.14 | 0.876 |
| $d(p_0 - \mu_0 + \bar\mu_1, p_1)$ | 6.002 | 6.083 | 68.50 | 0.929 | 6.222 | 6.346 | 74.90 | 0.876 |
| $d(p_0 - \mu_0 + \mu_1, p_1)$ | 3.887 | 3.962 | 1.76 | 0.929 | – | – | – | – |
| FM | $3.985 \pm 0.054$ | $4.115 \pm 0.067$ | $4.64 \pm 0.43$ | $0.938 \pm 0.014$ | $4.340 \pm 0.078$ | $4.564 \pm 0.111$ | $13.00 \pm 0.67$ | $0.865 \pm 0.034$ |
| CGFM | $\mathbf{3.882 \pm 0.019}$ | $3.999 \pm 0.020$ | $3.16 \pm 0.89$ | $0.952 \pm 0.004$ | $4.443 \pm 0.033$ | $4.621 \pm 0.041$ | $17.00 \pm 1.03$ | $0.899 \pm 0.008$ |
| MFM ($k = 0$) | $3.905 \pm 0.005$ | $4.012 \pm 0.006$ | $4.18 \pm 0.25$ | $\mathbf{0.958 \pm 0.001}$ | $4.209 \pm 0.007$ | $4.380 \pm 0.012$ | $12.34 \pm 0.50$ | $\mathbf{0.918 \pm 0.002}$ |
| MFM ($k = 10$) | $3.896 \pm 0.033$ | $4.005 \pm 0.036$ | $3.89 \pm 0.44$ | $0.957 \pm 0.005$ | $4.216 \pm 0.090$ | $4.395 \pm 0.098$ | $11.99 \pm 2.36$ | $0.917 \pm 0.005$ |
| MFM ($k = 50$) | $3.902 \pm 0.018$ | $4.008 \pm 0.022$ | $4.20 \pm 0.17$ | $\mathbf{0.958 \pm 0.000}$ | $4.214 \pm 0.017$ | $4.396 \pm 0.020$ | $12.09 \pm 0.75$ | $0.916 \pm 0.002$ |
| MFM ($k = 100$) | $3.884 \pm 0.039$ | $\mathbf{3.986 \pm 0.044}$ | $3.77 \pm 0.49$ | $0.955 \pm 0.001$ | $\mathbf{4.100 \pm 0.093}$ | $\mathbf{4.269 \pm 0.104}$ | $\mathbf{8.96 \pm 1.88}$ | $0.917 \pm 0.004$ |
| FM$^{w/}\mathcal{N}$ | $6.892 \pm 0.027$ | $7.164 \pm 0.033$ | $57.03 \pm 1.00$ | $0.655 \pm 0.003$ | $7.114 \pm 0.100$ | $7.404 \pm 0.086$ | $64.97 \pm 3.79$ | $0.613 \pm 0.008$ |
| CGFM$^{w/}\mathcal{N}$ | $4.313 \pm 0.077$ | $4.480 \pm 0.081$ | $11.51 \pm 1.96$ | $0.918 \pm 0.004$ | $7.135 \pm 0.045$ | $7.390 \pm 0.037$ | $79.78 \pm 4.67$ | $0.637 \pm 0.010$ |
| MFM$^{w/}\mathcal{N}$ ($k = 0$) | $3.982 \pm 0.010$ | $4.095 \pm 0.015$ | $5.04 \pm 0.36$ | $0.951 \pm 0.002$ | $4.177 \pm 0.042$ | $4.355 \pm 0.048$ | $10.53 \pm 0.59$ | $0.911 \pm 0.001$ |
| MFM$^{w/}\mathcal{N}$ ($k = 10$) | $4.006 \pm 0.008$ | $4.119 \pm 0.012$ | $5.13 \pm 0.30$ | $0.948 \pm 0.001$ | $4.156 \pm 0.065$ | $4.324 \pm 0.067$ | $9.58 \pm 1.63$ | $0.912 \pm 0.003$ |
| MFM$^{w/}\mathcal{N}$ ($k = 50$) | $3.982 \pm 0.018$ | $4.095 \pm 0.016$ | $4.74 \pm 0.21$ | $0.951 \pm 0.002$ | $4.153 \pm 0.069$ | $4.324 \pm 0.070$ | $9.63 \pm 1.45$ | $0.912 \pm 0.002$ |
| MFM$^{w/}\mathcal{N}$ ($k = 100$) | $4.004 \pm 0.012$ | $4.119 \pm 0.014$ | $5.19 \pm 0.43$ | $0.949 \pm 0.002$ | $4.166 \pm 0.001$ | $4.341 \pm 0.003$ | $9.52 \pm 0.33$ | $0.915 \pm 0.005$ |
| ICNN | $4.308 \pm 0.034$ | $4.413 \pm 0.036$ | $7.07 \pm 0.13$ | $0.929 \pm 0.006$ | $4.488 \pm 0.035$ | $4.665 \pm 0.038$ | $17.60 \pm 0.55$ | $0.884 \pm 0.002$ |

Table 7: **Experimental results on the organoid drug-screen dataset** for population prediction of treatment response across left-out *patient* populations. Results shown in this table are for **dim = 43**. We report the mean and standard deviations across metrics computed over 3 patient splits. We denote the best non-OT method with **bold** and the best OT-based method with underline.

| | Train | | | | Test | | | |
|---|---|---|---|---|---|---|---|---|
| | $\mathcal{W}_1(\downarrow)$ | $\mathcal{W}_2(\downarrow)$ | MMD $_{(\times 10^{-3})}(\downarrow)$ | $r^2(\uparrow)$ | $\mathcal{W}_1(\downarrow)$ | $\mathcal{W}_2(\downarrow)$ | MMD $_{(\times 10^{-3})}(\downarrow)$ | $r^2(\uparrow)$ |
| $d(p_0, p_1)$ | $4.128 \pm 0.069$ | $4.254 \pm 0.076$ | $9.44 \pm 1.21$ | $0.937 \pm 0.007$ | $4.175 \pm 0.135$ | $4.303 \pm 0.174$ | $12.11 \pm 2.07$ | $0.902 \pm 0.006$ |
| $d(p_0 - \mu_0 + \bar\mu_1, p_1)$ | $6.007 \pm 0.027$ | $6.085 \pm 0.034$ | $68.39 \pm 0.99$ | $0.937 \pm 0.007$ | $6.158 \pm 0.239$ | $6.235 \pm 0.229$ | $77.74 \pm 10.72$ | $0.902 \pm 0.006$ |
| $d(p_0 - \mu_0 + \mu_1, p_1)$ | $3.905 \pm 0.042$ | $3.987 \pm 0.042$ | $2.01 \pm 0.22$ | $0.937 \pm 0.007$ | – | – | – | – |
| FM | $3.950 \pm 0.055$ | $4.087 \pm 0.056$ | $4.62 \pm 1.44$ | $0.936 \pm 0.006$ | $4.171 \pm 0.107$ | $4.315 \pm 0.142$ | $10.95 \pm 1.98$ | $0.897 \pm 0.023$ |
| CGFM | $3.868 \pm 0.034$ | $3.994 \pm 0.037$ | $3.61 \pm 0.40$ | $0.955 \pm 0.009$ | $4.189 \pm 0.088$ | $4.321 \pm 0.119$ | $11.57 \pm 0.96$ | $0.914 \pm 0.008$ |
| MFM ($k = 0$) | $3.860 \pm 0.064$ | $3.973 \pm 0.076$ | $\mathbf{3.50 \pm 0.87}$ | $0.966 \pm 0.005$ | $4.135 \pm 0.094$ | $4.268 \pm 0.128$ | $10.18 \pm 1.28$ | $0.918 \pm 0.007$ |
| MFM ($k = 10$) | $\mathbf{3.853 \pm 0.062}$ | $\mathbf{3.963 \pm 0.067}$ | $3.55 \pm 0.92$ | $\mathbf{0.968 \pm 0.005}$ | $4.112 \pm 0.086$ | $4.243 \pm 0.121$ | $9.90 \pm 0.99$ | $0.925 \pm 0.008$ |
| MFM ($k = 50$) | $3.863 \pm 0.042$ | $3.980 \pm 0.053$ | $3.64 \pm 0.64$ | $0.966 \pm 0.006$ | $\mathbf{4.087 \pm 0.122}$ | $\mathbf{4.218 \pm 0.160}$ | $\mathbf{9.26 \pm 1.56}$ | $0.926 \pm 0.007$ |
| MFM ($k = 100$) | $3.876 \pm 0.055$ | $3.990 \pm 0.062$ | $3.65 \pm 0.91$ | $0.965 \pm 0.004$ | $4.112 \pm 0.148$ | $4.244 \pm 0.186$ | $9.63 \pm 2.08$ | $\mathbf{0.931 \pm 0.002}$ |
| FM$^{w/}$OT | $3.866 \pm 0.056$ | $3.981 \pm 0.064$ | $3.76 \pm 1.02$ | $0.963 \pm 0.007$ | $\underline{4.064 \pm 0.152}$ | $\underline{4.189 \pm 0.194}$ | $\underline{9.44 \pm 2.49}$ | $\underline{0.932 \pm 0.005}$ |
| CGFM$^{w/}$OT | $\underline{3.763 \pm 0.049}$ | $\underline{3.866 \pm 0.057}$ | $\underline{2.38 \pm 0.59}$ | $\underline{0.974 \pm 0.004}$ | $4.087 \pm 0.129$ | $4.217 \pm 0.165$ | $9.83 \pm 2.03$ | $0.924 \pm 0.009$ |
| ICNN | $4.394 \pm 0.477$ | $4.508 \pm 0.518$ | $7.80 \pm 3.37$ | $0.914 \pm 0.092$ | $4.157 \pm 0.168$ | $4.282 \pm 0.213$ | $11.18 \pm 2.51$ | $0.904 \pm 0.005$ |

Table 8: **Extended experimental results on the organoid drug-screen dataset** for population prediction of treatment response across *replicas-1* populations. Results shown in this table are for **dim = 43**. Here we show results for the individual cell cultures: **Fibroblasts**, patient derived organoids (**PDOs**), and patient derived organoids with Fibroblasts (**PDOFs**), respectively. We consider 4 settings for MFM with varying nearest-neighbours parameter ($k = 0, 10, 50, 100$).

**Fibroblasts**

|  | Train | | | | Test | | | |
|---|---|---|---|---|---|---|---|---|
|  | $\mathcal{W}_1(\downarrow)$ | $\mathcal{W}_2(\downarrow)$ | MMD $_{(\times 10^{-3})}(\downarrow)$ | $r^2(\uparrow)$ | $\mathcal{W}_1(\downarrow)$ | $\mathcal{W}_2(\downarrow)$ | MMD $_{(\times 10^{-3})}(\downarrow)$ | $r^2(\uparrow)$ |
| FM | $3.748 \pm 0.030$ | $3.838 \pm 0.047$ | $2.17 \pm 0.19$ | $0.937 \pm 0.020$ | $3.730 \pm 0.018$ | $3.816 \pm 0.031$ | $2.29 \pm 0.11$ | $0.948 \pm 0.014$ |
| FM$^{w/}\mathcal{N}$ | $6.701 \pm 0.093$ | $7.045 \pm 0.094$ | $51.55 \pm 3.16$ | $0.621 \pm 0.007$ | $6.548 \pm 0.115$ | $6.885 \pm 0.113$ | $48.95 \pm 3.69$ | $0.647 \pm 0.007$ |
| CGFM | $\mathbf{3.620 \pm 0.004}$ | $\mathbf{3.674 \pm 0.004}$ | $\mathbf{0.86 \pm 0.07}$ | $0.979 \pm 0.001$ | $3.732 \pm 0.014$ | $3.797 \pm 0.015$ | $3.26 \pm 0.07$ | $0.959 \pm 0.004$ |
| CGFM$^{w/}\mathcal{N}$ | $3.671 \pm 0.011$ | $3.725 \pm 0.011$ | $0.96 \pm 0.07$ | $\mathbf{0.980 \pm 0.001}$ | $5.844 \pm 0.192$ | $6.148 \pm 0.204$ | $39.80 \pm 3.07$ | $0.695 \pm 0.007$ |
| ICNN | $4.075 \pm 0.009$ | $4.145 \pm 0.008$ | $4.47 \pm 0.04$ | $0.925 \pm 0.006$ | $3.784 \pm 0.007$ | $3.848 \pm 0.007$ | $3.95 \pm 0.07$ | $0.952 \pm 0.005$ |
| MFM$^{w/}\mathcal{N}$ ($k=0$) | $3.745 \pm 0.034$ | $3.808 \pm 0.045$ | $1.73 \pm 0.27$ | $0.968 \pm 0.010$ | $3.730 \pm 0.036$ | $3.792 \pm 0.045$ | $2.02 \pm 0.29$ | $0.967 \pm 0.011$ |
| MFM$^{w/}\mathcal{N}$ ($k=10$) | $3.718 \pm 0.023$ | $3.774 \pm 0.023$ | $1.48 \pm 0.22$ | $0.976 \pm 0.002$ | $3.719 \pm 0.023$ | $3.777 \pm 0.023$ | $2.15 \pm 0.25$ | $0.973 \pm 0.000$ |
| MFM$^{w/}\mathcal{N}$ ($k=50$) | $3.716 \pm 0.027$ | $3.771 \pm 0.027$ | $1.55 \pm 0.32$ | $0.976 \pm 0.001$ | $3.717 \pm 0.032$ | $3.773 \pm 0.031$ | $2.13 \pm 0.45$ | $0.976 \pm 0.001$ |
| MFM$^{w/}\mathcal{N}$ ($k=100$) | $3.712 \pm 0.013$ | $3.767 \pm 0.012$ | $1.37 \pm 0.17$ | $0.977 \pm 0.002$ | $3.708 \pm 0.011$ | $3.764 \pm 0.011$ | $1.81 \pm 0.29$ | $0.975 \pm 0.002$ |
| MFM ($k=0$) | $3.667 \pm 0.027$ | $3.722 \pm 0.028$ | $1.43 \pm 0.18$ | $0.975 \pm 0.002$ | $\mathbf{3.661 \pm 0.027}$ | $3.719 \pm 0.027$ | $\mathbf{1.74 \pm 0.08}$ | $\mathbf{0.976 \pm 0.001}$ |
| MFM ($k=10$) | $3.679 \pm 0.032$ | $3.734 \pm 0.032$ | $1.69 \pm 0.34$ | $0.974 \pm 0.001$ | $3.680 \pm 0.030$ | $3.738 \pm 0.030$ | $2.04 \pm 0.28$ | $0.975 \pm 0.001$ |
| MFM ($k=50$) | $3.664 \pm 0.042$ | $3.720 \pm 0.043$ | $1.46 \pm 0.25$ | $0.974 \pm 0.002$ | $3.664 \pm 0.039$ | $\mathbf{3.722 \pm 0.041}$ | $1.84 \pm 0.23$ | $0.975 \pm 0.001$ |
| MFM ($k=100$) | $3.676 \pm 0.005$ | $3.732 \pm 0.006$ | $1.56 \pm 0.27$ | $0.972 \pm 0.003$ | $3.674 \pm 0.001$ | $3.731 \pm 0.002$ | $2.05 \pm 0.20$ | $\mathbf{0.976 \pm 0.001}$ |

**PDOs**

|  | Train | | | | Test | | | |
|---|---|---|---|---|---|---|---|---|
|  | $\mathcal{W}_1(\downarrow)$ | $\mathcal{W}_2(\downarrow)$ | MMD $_{(\times 10^{-3})}(\downarrow)$ | $r^2(\uparrow)$ | $\mathcal{W}_1(\downarrow)$ | $\mathcal{W}_2(\downarrow)$ | MMD $_{(\times 10^{-3})}(\downarrow)$ | $r^2(\uparrow)$ |
| FM | $3.910 \pm 0.037$ | $3.998 \pm 0.042$ | $3.37 \pm 0.25$ | $0.964 \pm 0.009$ | $3.825 \pm 0.062$ | $3.937 \pm 0.072$ | $3.31 \pm 0.34$ | $0.965 \pm 0.012$ |
| FM$^{w/}\mathcal{N}$ | $7.334 \pm 0.101$ | $7.578 \pm 0.103$ | $69.40 \pm 3.18$ | $0.585 \pm 0.009$ | $7.461 \pm 0.091$ | $7.692 \pm 0.094$ | $72.49 \pm 3.39$ | $0.592 \pm 0.009$ |
| CGFM | $3.792 \pm 0.061$ | $3.866 \pm 0.063$ | $2.27 \pm 0.59$ | $0.979 \pm 0.003$ | $4.062 \pm 0.096$ | $4.181 \pm 0.103$ | $7.75 \pm 1.45$ | $0.950 \pm 0.011$ |
| CGFM$^{w/}\mathcal{N}$ | $\mathbf{3.746 \pm 0.032}$ | $\mathbf{3.807 \pm 0.037}$ | $1.19 \pm 0.29$ | $\mathbf{0.986 \pm 0.002}$ | $7.672 \pm 0.149$ | $7.909 \pm 0.144$ | $94.96 \pm 7.33$ | $0.602 \pm 0.022$ |
| ICNN | $4.533 \pm 0.008$ | $4.635 \pm 0.007$ | $11.53 \pm 0.11$ | $0.901 \pm 0.005$ | $4.152 \pm 0.014$ | $4.261 \pm 0.013$ | $8.66 \pm 0.27$ | $0.928 \pm 0.004$ |
| MFM$^{w/}\mathcal{N}$ ($k=0$) | $3.788 \pm 0.047$ | $3.851 \pm 0.053$ | $1.59 \pm 0.20$ | $0.982 \pm 0.002$ | $3.741 \pm 0.042$ | $3.829 \pm 0.047$ | $2.39 \pm 0.23$ | $0.978 \pm 0.002$ |
| MFM$^{w/}\mathcal{N}$ ($k=10$) | $3.822 \pm 0.063$ | $3.891 \pm 0.071$ | $1.76 \pm 0.54$ | $0.983 \pm 0.001$ | $3.785 \pm 0.028$ | $3.875 \pm 0.029$ | $2.82 \pm 0.49$ | $0.979 \pm 0.002$ |
| MFM$^{w/}\mathcal{N}$ ($k=50$) | $3.794 \pm 0.033$ | $3.858 \pm 0.034$ | $1.71 \pm 0.50$ | $0.985 \pm 0.001$ | $3.775 \pm 0.022$ | $3.868 \pm 0.020$ | $2.92 \pm 0.41$ | $0.980 \pm 0.002$ |
| MFM$^{w/}\mathcal{N}$ ($k=100$) | $3.783 \pm 0.018$ | $3.845 \pm 0.018$ | $1.53 \pm 0.19$ | $0.984 \pm 0.002$ | $\mathbf{3.749 \pm 0.027}$ | $\mathbf{3.835 \pm 0.028}$ | $2.59 \pm 0.24$ | $\mathbf{0.981 \pm 0.002}$ |
| MFM ($k=0$) | $3.843 \pm 0.056$ | $3.916 \pm 0.066$ | $2.65 \pm 0.81$ | $0.971 \pm 0.002$ | $3.794 \pm 0.065$ | $3.891 \pm 0.074$ | $3.34 \pm 1.03$ | $0.974 \pm 0.002$ |
| MFM ($k=10$) | $3.852 \pm 0.039$ | $3.932 \pm 0.045$ | $2.80 \pm 0.19$ | $0.972 \pm 0.003$ | $3.781 \pm 0.013$ | $3.879 \pm 0.015$ | $3.45 \pm 0.52$ | $0.978 \pm 0.002$ |
| MFM ($k=50$) | $3.844 \pm 0.036$ | $3.924 \pm 0.033$ | $2.51 \pm 0.01$ | $0.973 \pm 0.006$ | $3.791 \pm 0.029$ | $3.889 \pm 0.026$ | $3.33 \pm 0.14$ | $0.974 \pm 0.006$ |
| MFM ($k=100$) | $3.905 \pm 0.082$ | $3.988 \pm 0.088$ | $3.87 \pm 1.81$ | $0.974 \pm 0.001$ | $3.783 \pm 0.025$ | $3.882 \pm 0.029$ | $3.33 \pm 0.45$ | $0.976 \pm 0.003$ |

**PDOFs**

|  | Train | | | | Test | | | |
|---|---|---|---|---|---|---|---|---|
|  | $\mathcal{W}_1(\downarrow)$ | $\mathcal{W}_2(\downarrow)$ | MMD $_{(\times 10^{-3})}(\downarrow)$ | $r^2(\uparrow)$ | $\mathcal{W}_1(\downarrow)$ | $\mathcal{W}_2(\downarrow)$ | MMD $_{(\times 10^{-3})}(\downarrow)$ | $r^2(\uparrow)$ |
| FM | $4.117 \pm 0.006$ | $4.287 \pm 0.007$ | $\mathbf{5.75 \pm 0.63}$ | $0.953 \pm 0.007$ | $4.328 \pm 0.057$ | $4.514 \pm 0.064$ | $12.11 \pm 0.76$ | $0.911 \pm 0.017$ |
| FM$^{w/}\mathcal{N}$ | $6.686 \pm 0.093$ | $6.921 \pm 0.095$ | $52.12 \pm 2.39$ | $0.722 \pm 0.006$ | $6.906 \pm 0.073$ | $7.154 \pm 0.073$ | $59.73 \pm 3.16$ | $0.686 \pm 0.009$ |
| CGFM | $4.180 \pm 0.130$ | $4.385 \pm 0.145$ | $6.35 \pm 2.13$ | $0.933 \pm 0.020$ | $4.467 \pm 0.080$ | $4.653 \pm 0.083$ | $15.51 \pm 0.93$ | $0.906 \pm 0.004$ |
| CGFM$^{w/}\mathcal{N}$ | $5.150 \pm 0.046$ | $5.499 \pm 0.049$ | $23.79 \pm 1.21$ | $0.843 \pm 0.005$ | $7.039 \pm 0.147$ | $7.285 \pm 0.153$ | $78.96 \pm 6.83$ | $0.700 \pm 0.020$ |
| ICNN | $4.434 \pm 0.006$ | $4.582 \pm 0.006$ | $5.69 \pm 0.05$ | $0.954 \pm 0.002$ | $4.552 \pm 0.005$ | $4.735 \pm 0.006$ | $17.02 \pm 0.22$ | $0.898 \pm 0.005$ |
| MFM$^{w/}\mathcal{N}$ ($k=0$) | $4.287 \pm 0.019$ | $4.482 \pm 0.015$ | $8.41 \pm 0.93$ | $0.928 \pm 0.011$ | $4.218 \pm 0.016$ | $4.385 \pm 0.016$ | $\mathbf{8.62 \pm 0.24}$ | $0.905 \pm 0.002$ |
| MFM$^{w/}\mathcal{N}$ ($k=10$) | $4.372 \pm 0.045$ | $4.582 \pm 0.048$ | $9.70 \pm 1.33$ | $0.923 \pm 0.007$ | $4.326 \pm 0.052$ | $4.501 \pm 0.057$ | $10.86 \pm 0.60$ | $0.904 \pm 0.004$ |
| MFM$^{w/}\mathcal{N}$ ($k=50$) | $4.356 \pm 0.036$ | $4.566 \pm 0.040$ | $9.72 \pm 0.68$ | $0.922 \pm 0.006$ | $4.291 \pm 0.014$ | $4.469 \pm 0.018$ | $10.14 \pm 0.60$ | $0.905 \pm 0.001$ |
| MFM$^{w/}\mathcal{N}$ ($k=100$) | $4.295 \pm 0.026$ | $4.495 \pm 0.033$ | $8.55 \pm 0.56$ | $0.932 \pm 0.007$ | $4.251 \pm 0.071$ | $4.423 \pm 0.080$ | $9.44 \pm 1.51$ | $0.905 \pm 0.003$ |
| MFM ($k=0$) | $\mathbf{4.111 \pm 0.016}$ | $\mathbf{4.281 \pm 0.023}$ | $6.04 \pm 0.26$ | $\mathbf{0.954 \pm 0.009}$ | $\mathbf{4.184 \pm 0.022}$ | $\mathbf{4.359 \pm 0.022}$ | $8.96 \pm 0.64$ | $0.915 \pm 0.003$ |
| MFM ($k=10$) | $4.158 \pm 0.028$ | $4.332 \pm 0.029$ | $6.97 \pm 0.80$ | $0.946 \pm 0.006$ | $4.235 \pm 0.019$ | $4.418 \pm 0.022$ | $9.89 \pm 0.81$ | $0.914 \pm 0.002$ |
| MFM ($k=50$) | $4.155 \pm 0.035$ | $4.330 \pm 0.015$ | $6.80 \pm 0.98$ | $0.943 \pm 0.002$ | $4.245 \pm 0.044$ | $4.429 \pm 0.035$ | $10.00 \pm 0.28$ | $0.912 \pm 0.001$ |
| MFM ($k=100$) | $4.137 \pm 0.048$ | $4.305 \pm 0.068$ | $6.73 \pm 0.42$ | $0.945 \pm 0.002$ | $4.236 \pm 0.001$ | $4.412 \pm 0.004$ | $10.18 \pm 0.10$ | $\mathbf{0.918 \pm 0.005}$ |

Table 9: **Extended experimental results on the organoid drug-screen dataset** for population prediction of treatment response across *replicas-2* populations. Results shown in this table are for **dim = 43**. Here we show results for the individual cell cultures: **Fibroblasts**, patient derived organoids (**PDOs**), and patient derived organoids with Fibroblasts (**PDOFs**), respectively. We consider 4 settings for MFM with varying nearest-neighbours parameter ($k = 0, 10, 50, 100$).

**Fibroblasts**

| | Train | | | | Test | | | |
|---|---|---|---|---|---|---|---|---|
| | $\mathcal{W}_1(\downarrow)$ | $\mathcal{W}_2(\downarrow)$ | MMD $_{(\times 10^{-3})}(\downarrow)$ | $r^2(\uparrow)$ | $\mathcal{W}_1(\downarrow)$ | $\mathcal{W}_2(\downarrow)$ | MMD $_{(\times 10^{-3})}(\downarrow)$ | $r^2(\uparrow)$ |
| FM | $3.694 \pm 0.018$ | $3.775 \pm 0.023$ | $1.74 \pm 0.20$ | $0.951 \pm 0.007$ | $3.646 \pm 0.163$ | $3.805 \pm 0.231$ | $3.67 \pm 1.46$ | $0.867 \pm 0.070$ |
| FM$^{w/}\mathcal{N}$ | $6.790 \pm 0.109$ | $7.124 \pm 0.120$ | $54.97 \pm 1.68$ | $0.661 \pm 0.005$ | $6.905 \pm 0.092$ | $7.252 \pm 0.100$ | $64.40 \pm 1.31$ | $0.607 \pm 0.012$ |
| CGFM | $\mathbf{3.616 \pm 0.031}$ | $\mathbf{3.671 \pm 0.032}$ | $\mathbf{0.90 \pm 0.13}$ | $0.980 \pm 0.003$ | $3.530 \pm 0.011$ | $3.584 \pm 0.010$ | $4.37 \pm 0.55$ | $0.975 \pm 0.003$ |
| CGFM$^{w/}\mathcal{N}$ | $3.663 \pm 0.011$ | $3.718 \pm 0.011$ | $0.93 \pm 0.13$ | $\mathbf{0.982 \pm 0.001}$ | $5.813 \pm 0.372$ | $6.136 \pm 0.398$ | $44.44 \pm 8.18$ | $0.688 \pm 0.026$ |
| ICNN | $4.054 \pm 0.03$ | $4.124 \pm 0.033$ | $4.43 \pm 0.14$ | $0.925 \pm 0.007$ | $3.477 \pm 0.031$ | $3.534 \pm 0.033$ | $3.67 \pm 0.17$ | $0.969 \pm 0.004$ |
| MFM$^{w/}\mathcal{N}$ ($k=0$) | $3.671 \pm 0.021$ | $3.726 \pm 0.022$ | $1.17 \pm 0.17$ | $0.979 \pm 0.001$ | $3.503 \pm 0.030$ | $3.555 \pm 0.031$ | $\mathbf{1.68 \pm 0.11}$ | $0.975 \pm 0.001$ |
| MFM$^{w/}\mathcal{N}$ ($k=10$) | $3.691 \pm 0.015$ | $3.748 \pm 0.014$ | $1.18 \pm 0.14$ | $0.978 \pm 0.000$ | $3.526 \pm 0.024$ | $3.582 \pm 0.022$ | $1.86 \pm 0.36$ | $0.972 \pm 0.001$ |
| MFM$^{w/}\mathcal{N}$ ($k=50$) | $3.678 \pm 0.017$ | $3.733 \pm 0.017$ | $1.15 \pm 0.12$ | $0.979 \pm 0.001$ | $3.515 \pm 0.029$ | $3.567 \pm 0.029$ | $2.07 \pm 0.63$ | $0.977 \pm 0.002$ |
| MFM$^{w/}\mathcal{N}$ ($k=100$) | $3.706 \pm 0.020$ | $3.764 \pm 0.022$ | $1.44 \pm 0.08$ | $0.977 \pm 0.001$ | $3.536 \pm 0.036$ | $3.590 \pm 0.037$ | $1.90 \pm 0.23$ | $0.975 \pm 0.003$ |
| MFM ($k=0$) | $3.626 \pm 0.012$ | $3.682 \pm 0.013$ | $1.20 \pm 0.04$ | $0.979 \pm 0.000$ | $3.451 \pm 0.020$ | $3.505 \pm 0.021$ | $2.55 \pm 0.42$ | $0.981 \pm 0.002$ |
| MFM ($k=10$) | $3.624 \pm 0.025$ | $3.678 \pm 0.026$ | $1.04 \pm 0.17$ | $0.981 \pm 0.000$ | $3.451 \pm 0.037$ | $3.504 \pm 0.038$ | $2.47 \pm 0.56$ | $\mathbf{0.982 \pm 0.001}$ |
| MFM ($k=50$) | $3.639 \pm 0.016$ | $3.694 \pm 0.016$ | $1.71 \pm 0.20$ | $0.979 \pm 0.001$ | $\mathbf{3.443 \pm 0.035}$ | $\mathbf{3.497 \pm 0.037}$ | $2.89 \pm 1.85$ | $0.981 \pm 0.001$ |
| MFM ($k=100$) | $3.654 \pm 0.005$ | $3.712 \pm 0.005$ | $1.56 \pm 0.20$ | $0.978 \pm 0.000$ | $3.480 \pm 0.027$ | $3.534 \pm 0.029$ | $3.32 \pm 0.49$ | $0.980 \pm 0.001$ |

**PDOs**

| | Train | | | | Test | | | |
|---|---|---|---|---|---|---|---|---|
| | $\mathcal{W}_1(\downarrow)$ | $\mathcal{W}_2(\downarrow)$ | MMD $_{(\times 10^{-3})}(\downarrow)$ | $r^2(\uparrow)$ | $\mathcal{W}_1(\downarrow)$ | $\mathcal{W}_2(\downarrow)$ | MMD $_{(\times 10^{-3})}(\downarrow)$ | $r^2(\uparrow)$ |
| FM | $3.948 \pm 0.092$ | $4.050 \pm 0.116$ | $3.96 \pm 0.68$ | $0.942 \pm 0.023$ | $4.043 \pm 0.100$ | $4.174 \pm 0.131$ | $4.90 \pm 0.50$ | $0.931 \pm 0.033$ |
| FM$^{w/}\mathcal{N}$ | $7.109 \pm 0.028$ | $7.351 \pm 0.030$ | $63.42 \pm 1.97$ | $0.626 \pm 0.008$ | $7.697 \pm 0.212$ | $7.976 \pm 0.182$ | $77.52 \pm 7.33$ | $0.519 \pm 0.002$ |
| CGFM | $3.750 \pm 0.013$ | $3.815 \pm 0.017$ | $1.77 \pm 0.71$ | $0.977 \pm 0.004$ | $4.225 \pm 0.146$ | $4.333 \pm 0.171$ | $9.50 \pm 3.14$ | $0.933 \pm 0.016$ |
| CGFM$^{w/}\mathcal{N}$ | $\mathbf{3.728 \pm 0.024}$ | $\mathbf{3.787 \pm 0.026}$ | $\mathbf{1.14 \pm 0.16}$ | $\mathbf{0.985 \pm 0.000}$ | $8.333 \pm 0.309$ | $8.534 \pm 0.298$ | $114.45 \pm 14.17$ | $0.508 \pm 0.016$ |
| ICNN | $4.476 \pm 0.039$ | $4.58 \pm 0.041$ | $11.17 \pm 0.05$ | $0.906 \pm 0.007$ | $4.413 \pm 0.048$ | $4.519 \pm 0.051$ | $11.18 \pm 0.05$ | $0.886 \pm 0.004$ |
| MFM$^{w/}\mathcal{N}$ ($k=0$) | $3.775 \pm 0.012$ | $3.837 \pm 0.013$ | $1.75 \pm 0.05$ | $0.979 \pm 0.001$ | $\mathbf{3.855 \pm 0.075}$ | $3.944 \pm 0.087$ | $3.73 \pm 0.79$ | $0.978 \pm 0.003$ |
| MFM$^{w/}\mathcal{N}$ ($k=10$) | $3.799 \pm 0.009$ | $3.860 \pm 0.009$ | $1.89 \pm 0.26$ | $0.978 \pm 0.001$ | $3.877 \pm 0.042$ | $3.961 \pm 0.047$ | $3.53 \pm 0.30$ | $0.978 \pm 0.003$ |
| MFM$^{w/}\mathcal{N}$ ($k=50$) | $3.800 \pm 0.023$ | $3.863 \pm 0.023$ | $1.93 \pm 0.27$ | $0.980 \pm 0.001$ | $3.857 \pm 0.059$ | $\mathbf{3.938 \pm 0.066}$ | $\mathbf{3.40 \pm 0.77}$ | $\mathbf{0.981 \pm 0.001}$ |
| MFM$^{w/}\mathcal{N}$ ($k=100$) | $3.800 \pm 0.008$ | $3.863 \pm 0.010$ | $2.05 \pm 0.12$ | $0.979 \pm 0.001$ | $3.882 \pm 0.008$ | $3.973 \pm 0.009$ | $3.80 \pm 0.26$ | $0.979 \pm 0.000$ |
| MFM ($k=0$) | $3.797 \pm 0.020$ | $3.864 \pm 0.020$ | $2.40 \pm 0.32$ | $0.971 \pm 0.001$ | $3.894 \pm 0.026$ | $3.982 \pm 0.037$ | $3.87 \pm 0.83$ | $0.976 \pm 0.002$ |
| MFM ($k=10$) | $3.792 \pm 0.041$ | $3.863 \pm 0.049$ | $2.29 \pm 0.47$ | $0.975 \pm 0.005$ | $3.953 \pm 0.095$ | $4.055 \pm 0.115$ | $4.80 \pm 2.00$ | $0.978 \pm 0.001$ |
| MFM ($k=50$) | $3.796 \pm 0.014$ | $3.864 \pm 0.016$ | $2.50 \pm 0.22$ | $0.972 \pm 0.001$ | $4.035 \pm 0.099$ | $4.153 \pm 0.116$ | $6.40 \pm 2.19$ | $0.971 \pm 0.004$ |
| MFM ($k=100$) | $3.798 \pm 0.037$ | $3.866 \pm 0.041$ | $2.54 \pm 0.28$ | $0.970 \pm 0.002$ | $3.869 \pm 0.094$ | $3.953 \pm 0.113$ | $3.49 \pm 0.76$ | $0.975 \pm 0.001$ |

**PDOFs**

| | Train | | | | Test | | | |
|---|---|---|---|---|---|---|---|---|
| | $\mathcal{W}_1(\downarrow)$ | $\mathcal{W}_2(\downarrow)$ | MMD $_{(\times 10^{-3})}(\downarrow)$ | $r^2(\uparrow)$ | $\mathcal{W}_1(\downarrow)$ | $\mathcal{W}_2(\downarrow)$ | MMD $_{(\times 10^{-3})}(\downarrow)$ | $r^2(\uparrow)$ |
| FM | $4.313 \pm 0.062$ | $4.521 \pm 0.076$ | $8.21 \pm 0.55$ | $0.919 \pm 0.014$ | $5.330 \pm 0.112$ | $5.712 \pm 0.113$ | $30.45 \pm 3.30$ | $\mathbf{0.798 \pm 0.002}$ |
| FM$^{w/}\mathcal{N}$ | $6.778 \pm 0.046$ | $7.015 \pm 0.042$ | $52.69 \pm 2.63$ | $0.708 \pm 0.004$ | $6.741 \pm 0.177$ | $6.985 \pm 0.163$ | $52.98 \pm 5.35$ | $0.713 \pm 0.014$ |
| CGFM | $4.281 \pm 0.065$ | $4.511 \pm 0.078$ | $\mathbf{6.81 \pm 1.33}$ | $0.898 \pm 0.011$ | $5.574 \pm 0.051$ | $5.945 \pm 0.052$ | $37.13 \pm 0.40$ | $0.790 \pm 0.009$ |
| CGFM$^{w/}\mathcal{N}$ | $5.549 \pm 0.206$ | $5.936 \pm 0.214$ | $32.44 \pm 5.58$ | $0.787 \pm 0.014$ | $7.258 \pm 0.197$ | $7.501 \pm 0.212$ | $80.45 \pm 8.02$ | $0.713 \pm 0.005$ |
| ICNN | $4.393 \pm 0.034$ | $4.534 \pm 0.037$ | $5.60 \pm 0.20$ | $\mathbf{0.956 \pm 0.003}$ | $5.573 \pm 0.049$ | $5.943 \pm 0.056$ | $37.96 \pm 1.58$ | $\mathbf{0.798 \pm 0.004}$ |
| MFM$^{w/}\mathcal{N}$ ($k=0$) | $4.500 \pm 0.039$ | $4.721 \pm 0.043$ | $12.21 \pm 1.07$ | $0.896 \pm 0.005$ | $5.173 \pm 0.078$ | $5.567 \pm 0.077$ | $26.17 \pm 2.29$ | $0.779 \pm 0.003$ |
| MFM$^{w/}\mathcal{N}$ ($k=10$) | $4.528 \pm 0.033$ | $4.751 \pm 0.045$ | $12.31 \pm 0.73$ | $0.888 \pm 0.003$ | $5.065 \pm 0.169$ | $5.431 \pm 0.172$ | $23.34 \pm 4.74$ | $0.786 \pm 0.005$ |
| MFM$^{w/}\mathcal{N}$ ($k=50$) | $4.468 \pm 0.026$ | $4.690 \pm 0.030$ | $11.15 \pm 0.34$ | $0.894 \pm 0.006$ | $5.088 \pm 0.121$ | $5.468 \pm 0.115$ | $23.43 \pm 3.15$ | $0.778 \pm 0.005$ |
| MFM$^{w/}\mathcal{N}$ ($k=100$) | $4.508 \pm 0.042$ | $4.731 \pm 0.042$ | $12.09 \pm 1.34$ | $0.891 \pm 0.006$ | $5.082 \pm 0.037$ | $5.461 \pm 0.047$ | $23.16 \pm 0.90$ | $0.792 \pm 0.015$ |
| MFM ($k=0$) | $4.293 \pm 0.005$ | $4.491 \pm 0.005$ | $8.94 \pm 0.54$ | $0.925 \pm 0.002$ | $5.283 \pm 0.017$ | $5.653 \pm 0.029$ | $30.60 \pm 0.77$ | $0.797 \pm 0.005$ |
| MFM ($k=10$) | $4.273 \pm 0.039$ | $4.474 \pm 0.041$ | $8.35 \pm 0.83$ | $0.915 \pm 0.009$ | $5.244 \pm 0.168$ | $5.626 \pm 0.175$ | $28.71 \pm 5.10$ | $0.790 \pm 0.016$ |
| MFM ($k=50$) | $\mathbf{4.271 \pm 0.038}$ | $4.466 \pm 0.049$ | $8.40 \pm 0.40$ | $0.924 \pm 0.001$ | $5.165 \pm 0.132$ | $5.538 \pm 0.140$ | $26.97 \pm 2.46$ | $0.796 \pm 0.009$ |
| MFM ($k=100$) | $4.200 \pm 0.079$ | $\mathbf{4.381 \pm 0.095}$ | $7.20 \pm 1.13$ | $0.918 \pm 0.004$ | $\mathbf{4.950 \pm 0.199}$ | $\mathbf{5.321 \pm 0.219}$ | $\mathbf{20.08 \pm 5.10}$ | $0.798 \pm 0.013$ |

## F.1 ANALYSIS OF POPULATION EMBEDDINGS

Here, we analyze the embedding space of the population embedding model $\varphi(p_0; \theta)$ for the synthetic letters dataset and the organoid drug-screen dataset. To do this, we compute embeddings $h = \varphi(\{x_0^j\}_{j=1}^{N'}; \theta)$ for each initial population. For letters we consider 200 random rotations per letter and for the organoid drug-screen dataset we consider all population pairs $(p_0, p_1)$. We then project the embeddings into a 2-dimensional space using uniform manifold approximation and projection (UMAP) to visualize model embeddings. We compute pairwise Euclidean distances of the samples in the projected embedding space and report these distances visually using heat maps. See Fig. 7.

On the letters dataset, the population embedding model learns to embed similar letter close silhouettes together. For example, 'A' and 'V', 'T' and 'Y' (even though 'Y' silhouettes are never seen during training), and more. Note, that in this dataset, letter silhouette populations $p_0$ are both corrupted with noise and rotated randomly. We conduct the same analysis for the organoid drug-screen dataset. We observe that in some sense $\varphi(p_0; \theta)$ groups populations from particular patients in a generally consistent matter to that found in Ramos Zapatero et al. (2023). Specifically, we observe patient who are chemosensitive (PDO 21, 75, 23, 27) cluster together, away from patients who are chemorefractory (PDO 5, 11, 141, 216). This illustrates the embedding are able to capture patient drug response characteristics from untreated patient sample $p_0$.

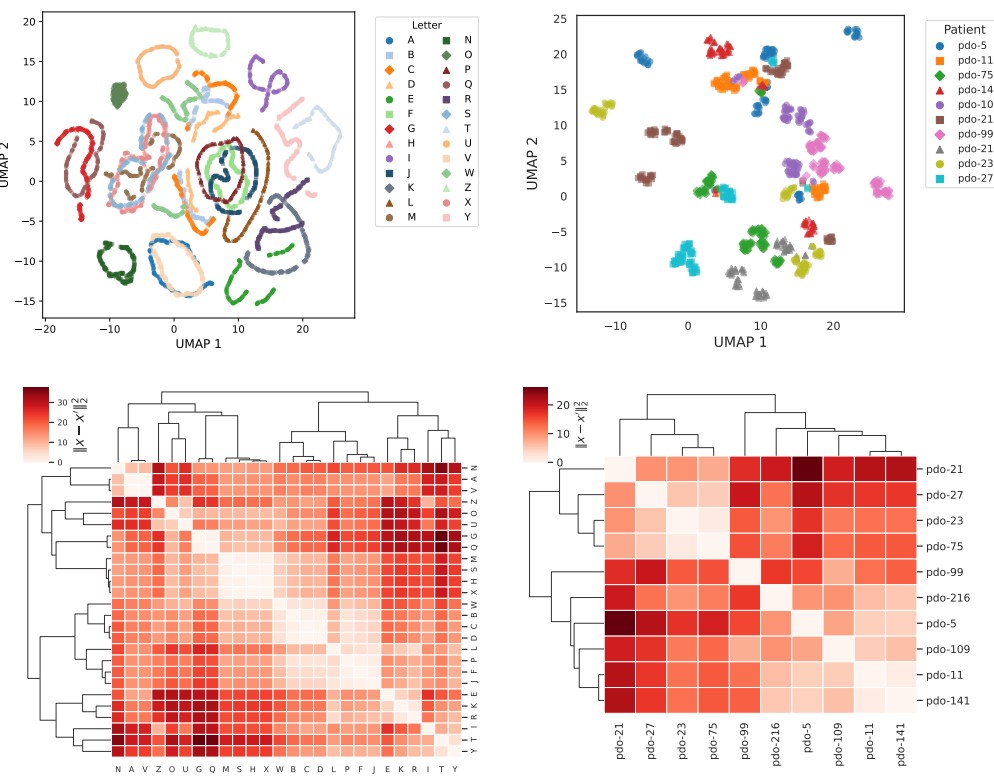

Figure 7: **Analysis of population embeddings from $\varphi(p_0; \theta)$.** We provide a UMAP visualization of population embeddings space (top) and pairwise distances of population embeddings computed in 2-dimensional space (bottom). (*Left*) analysis of letter population embeddings plotted for 200 random rotations of each noisy letter silhouette. (*Right*) analysis of patient population embeddings plotted for all control populations.

## F.2 ANALYSIS OF PREDICTION QUALITY ON ORGANOID DRUG-SCREEN DATA

In this section, we analyze the predictions produced by vector field models trained with MFM. For this analysis, we look at model predictions for the three respective left-out test patient splits: PDO-21, PDO-27, PDO-75. For each control population in the patient split (specifically, for the test patient in the given split), we predict the respective treatment response. We then subset the data to look

at populations in the PDOF culture and take the mean of observations in each population (for both predicted and target populations). We project the means of the predicted and target populations into 2D space using principal component analysis (PCA) across all. We fit PCA using means from source, target, and predicted samples. Lastly, we plot the target and predicted samples separately. We show the result of this analysis in Fig. 8.

We observe that for all three test patients the general structure is preserved, with the treatment, Oxaliplatin (Green), being the furthest away both for the target and ground truth datasets. This is because (as shown in the original paper by Ramos Zapatero et al. (2023)) Oxaliplatin has a large effect on these cancer cells for the PDOF subset of PDOs. We see from Fig. 8 that this is more pronounced for PDO-21 and PDO-27 than for PDO-75. Overall, this reflects the conclusions drawn from Figure 4 in the original dataset paper Ramos Zapatero et al. (2023). In this way, we are able to draw similar conclusions from the predicted populations as from the data.

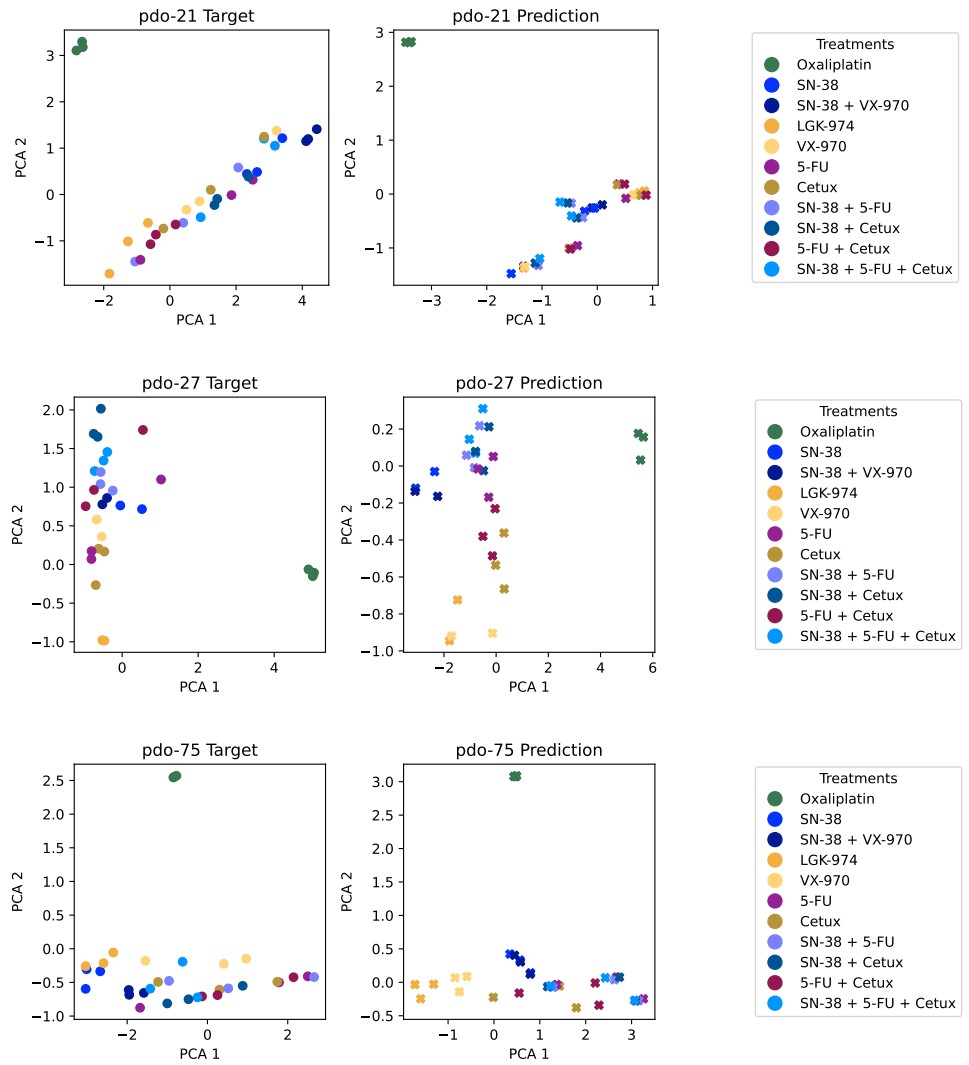

Figure 8: **Analysis of treatment-specific response prediction.** We plot population means in 2D PCA space for target populations (*Left*), predicted populations (*Middle*), and the respective treatment identifier legend (*Right*).

