# OpenReview forum: "Meta Flow Matching: Integrating Vector Fields on the Wasserstein Manifold"
_ICLR.cc/2025/Conference — ICLR 2025 Poster_

### Official Review · Reviewer_kKjh · 2024-10-27

**Soundness:** 3
**Presentation:** 2
**Contribution:** 3
**Rating:** 6
**Confidence:** 3

**Summary:**

- The paper proposes a new diagram of conditional flow matching models named meta flow matching.
- The high-level understanding is: meta flow matching represents the conditional input in a new way -- by embedding the information of the whole population (or saying distributions).
- In math, this can be formulated as constructing the vector field in the Wasserstein manifold.
- The authors then provide the technical construction of the model, including objective functions and parametrization.
- Numerical experiments are conducted on synthetic and organoid drug-screen datasets.
- I am happy to see this conceptual novelty in the approach.
- I didn't thoroughly check the technical part in the paper while I didn't see anything odd.
- I am a little concerned on certain claims and the effectiveness of the method, both conceptually and numerically.

**Strengths:**

I am listing both strengths (+), weakness (-) and questions (?) in the following to make the review more logistic.

- (+) The meta flow matching diagram is clearly novel. The specific new part is the population/distribution embedding that the model can distinguish and evolve differently in different population, for the same particle.
- (?) Question on ODE: Then what is the particle-level ODE corresponding to the PDE (12)? Is it similar to FP-equation that dx/dt = vt(x, pt)? Can the author prove it or give the detailed reference?
- (?) Can you method be extent to SDE case?
- (?) Analysis of population embeddings: To learn the population embeddings, I believe you should have enough example of populations otherwise the generalization is less promised. I would like to see a detailed analysis of population embeddings for a sanity check in both experiments.
- (-) Reduced number of data points: Following the last point, for the organoid drug-screen experiment where only contains 10 patients, does it mean there are only 10 populations (data point for patient population embeddings)? It is out of expectation that model can capture the right population embeddings from 10 data points.
- (?) Why replicate-split in the organoid drug-screen experiment is meaningful? In my understanding the conclusion drawn from different replicates usually would be consistent (that they has similar patterns).
- (?) I also do not quite understand why other baselines under-perform in the replicate-split since if I understand correctly, generalization would not be a big challenge here.
- (?) Can you provide a random baseline for your experiments? Random guess baseline and random initialized model baseline.
- (-) An extremely important reference of "Deep Generalized Schrödinger Bridge" is missed here. Conceptually this paper should be sth ahead of yours and need to be discussed in details. Population embedding is also adopted in the paper in a simpler way.
- (?) To further illustrate the effectviese, I think comparison with the numbers and experiments in "Deep Generalized Schrödinger Bridge" is necessary.
- (?) Comparison with the number and experiment in "Learning single-cell perturbation responses using neural optimal transport" is also necessary in my eye. This is a classical paper and one of your competitor method.
- (?) Complexity: Can you provide a complexity analysis here? Your method need additional embedding of the whole population and I think the scalability performance needed to be shown somehow (w.r.t. the size of population) which is very useful to the community for method selection.
- (?) How do you construct the graph in the organoid drug-screen experiment? How different graph construction impact your model results?

**Weaknesses:**

See Strengths.

**Questions:**

See Strengths.

---

> ### Author Response · Authors · 2024-11-21
> **(1/3)**
>
> We thank the reviewer for their time and effort in reviewing our paper. We are heartened to hear that the reviewer views that MFM is "clearly novel" and is "happy to see the conceptual novelty" of our method. We next answer all the important questions raised by the reviewer while refer to the general response to all reviewers for any additional experiments.
>
> >Question on ODE: Then what is the particle-level ODE corresponding to the PDE (12)? Is it similar to FP-equation that dx/dt = vt(x, pt)? Can the author prove it or give the detailed reference?
>
> The FP-equation is a special case of the PDE in equation (12) when, for instance, we take $v_t(x, p_t) = -\frac{1}{2}\nabla_x \log p_t(x)$, as we discuss in Example 2. In general, equation (12) is our _assumption_ on the dynamics of the system (for letters we know it is diffusion, for cells it is unknown). To be precise, we assume that the vector field $v_t(x, p_t)$ can be represented as a function of density $p_t$, and try to learn this functional dependency via a GNN. However, clearly, we do not know the analytic or any other form of how $v_t(x, p_t)$ depends on $p_t$.
>
> > Can you method be extent to SDE case?
>
> Yes. It is straightforward to extend MFM to the stochastic setting. We leave exploration of empirical performance of MFM in the stochastic setting to future work.
>
> >Analysis of population embeddings: To learn the population embeddings, I believe you should have enough example of populations otherwise the generalization is less promised.
>
> We agree that generalization is likely to improve with increasing data. We refer the reviewer to Figure 5 for an ablation over the number of letter populations and its effect on generalization performance in the synthetic letters setting. We indeed observe that as we increase the number of training populations, the generalization performance of MFM also improves. In contrast, the performance of the baseline methods, which are not designed to operate in the regime of generalizing to unseen distributions, does not meaningfully improve.
>
> >I would like to see a detailed analysis of population embeddings for a sanity check in both experiments.
>
> We thank the reviewer for their helpful suggestion. To illustrate the ability of MFM to learn meaningful population embeddings, we have added a detailed analysis of population embeddings both in the synthetic data setting and in the organoid drug-screen dataset in Appendix F.1. To do this, we first use Uniform Manifold Approximation and Projection (UMAP) to project embeddings in 2 dimensions. We then compute the pairwise distances between the distributions (letters silhouette populations for the synthetic data and control populations for organoid drug-screen data). Through this analysis, we found that the embeddings reflect data characteristics and clusters of similar groups. Please see Appendix F.1 and Figure 7 for details in the updated text. On the organoid drug-screen data, we found that cell populations from patients with similar chemotherapeutic drug responses also cluster together. Ramos Zapatero et al., (2023) identified patients who are chemosensitive (responsive to chemotherapy), chemorefractory (not responsive to chemotherapy). In turn, we observed PDO11 and PDO141 clustering together which are derived from different patients and both classified as chemorefractory.

---

> > ### Author Response · Authors · 2024-11-21
> > **(2/3)**
> >
> > >For the organoid drug-screen experiment where only contains 10 patients, does it mean there are only 10 populations (data point for patient population embeddings)? It is out of expectation that model can capture the right population embeddings from 10 data points.
> >
> > For the organoid drug-screen dataset, there are 10 patients with many replicated conditions/experiments, hence leading to a large quantity of $(p_0, p_1)$ population pairs to use for training and evaluation. In total, we have 927 $(p_0, p_1)$ population pairs, which we divide into training and testing splits. For the patient split, we leave out all $(p_0, p_1)$ pairs fully from 1 patient to evaluate how methods perform when predicting population dynamics in an unseen patient.
> >
> > >Why replicate-split in the organoid drug-screen experiment is meaningful? In my understanding the conclusion drawn from different replicates usually would be consistent (that they has similar patterns).
> >
> > Although the generalization task in the replicate split is easier, since as the reviewer reasonably pointed out, there should arguably be less diversity between populations in the train and test sets. However, there is still sufficient diversity between populations due to inherit biological heterogeneity that exists across cell populations. We can thus use this inherent biological heterogeneity across cells (which exists in the replica split) to pose a generalization problem across unseen populations and learn meaningful embeddings.
> >
> > >Can you provide a random baseline for your experiments? Random guess baseline and random initialized model baseline.
> >
> > Thank you for the suggestion. We have included two additional baselines. We use $d(p_0, p_1)$ to denote comparison of the unperturbed source distribution $p_0$ with the base target distribution $p_1$. We also consider $d(p_0 - \mu_0 + \tilde{\mu}_1, p_1)$ to denote the comparison between the base target distribution with the source distribution shifted by a constant vector $\tilde{\mu_1}$. We compute $\tilde{\mu_1}$ by taking the average of all $p_1$ population means in the training set. In Table 2 and Table 3, we see that all models perform better than these trivial baselines. We see that MFM is consistently outperforming these baselines. This further supports the fact that MFM is able to generalize across initial distributions relative to FM, CGFM, and ICNN baselines.
> >
> > >I also do not quite understand why other baselines under-perform in the replicate-split since if I understand correctly, generalization would not be a big challenge here.
> >
> > Re-iterating from the reviewer's previous question regarding the replica split and generalization. We agree that generalization is easier in replica split relative to the patient split. However, it is still not trivial to generalize across populations in the left-out replicas. This split tests the model's ability to generalize across the underlying biological heterogeneity of different cell populations. For reference, please refer to the updated Table 2, where we include two additional baselines on the replica split. Through these results, we observe that there exists sufficient biological diversity between experimental conditions such that models learn something meaningful, and in particular that MFM can better generalize relative to baseline.
> >
> > >An extremely important reference of "Deep Generalized Schrödinger Bridge" is missed here. Conceptually this paper should be sth ahead of yours and need to be discussed in details. Population embedding is also adopted in the paper in a simpler way. To further illustrate the effectviese, I think comparison with the numbers and experiments in "Deep Generalized Schrödinger Bridge" is necessary.
> >
> > We thank the reviewer for bringing this important reference to our attention. The work "Deep Generalized Schrödinger Bridge" (DeepGSB) also considers interacting terms between particles, specifically entropy and congestion but does not consider embeddings of populations. Furthermore, the focus of this work is generalization over initial distributions, which DeepGSB does not consider. For this reason, the experiments in DeepGSB are outside of the scope of this work. We add a discussion of DeepGSB in our related work section.

---

> > > ### Author Response · Authors · 2024-11-21
> > > **(3/3)**
> > >
> > > >Comparison with the number and experiment in "Learning single-cell perturbation responses using neural optimal transport" is also necessary in my eye. This is a classical paper and one of your competitor method.
> > >
> > > Thank you for this point. We would like to clarify that we in fact do compare with the method in "Learning single-cell perturbation responses using neural optimal transport" (CellOT, which we denote as the ICNN). Moreover, we point out that the experiments in CellOT consider a *single* control distribution. For this reason, the experiments in that work are not applicable to our setting, which considers generalization over *multiple* control distributions. We compare MFM to the CellOT model in Table 2 and Table 3 (updated) referred to by its underlying architecture an input convex neural network (ICNN). We have clarified this in the text.
> > >
> > > >Can you provide a complexity analysis here? Your method need additional embedding of the whole population and I think the scalability performance needed to be shown somehow (w.r.t. the size of population) which is very useful to the community for method selection.
> > >
> > > We thank the reviewer for asking these important questions. Complexity depends on the model architecture. For MFM, we use a graph convolutional network (GCN) with $k$-nearest neighbors edge pooling to construct particle interaction graphs (lines 369-371). Here, as $k$ increases, training time and memory usage also increase. There are other approaches to do this, but the focus of this work is in the general idea, theory, and execution of learning population/distribution embeddings and generalizing across unseen populations. We leave exploring improvements in this regard for future work.
> > >
> > > >How do you construct the graph in the organoid drug-screen experiment? How different graph construction impact your model results?
> > >
> > > We do not explicitly construct a particle interaction graph. Rather, we let the population embedding model handle interactions for us -- i.e., using knn edge pooling layers in GCN model. Explicit modeling of the particle interactions is in general a very challenging problem. We leave an investigation of explicit modeling of the particle interactions for future work.
> > >
> > > We once again appreciate your time and effort in this rebuttal period. We believe we have addressed the concerns brought up by the reviewer, and through the reviewer's insightful suggestions, we have improved the overall quality of our work. If the reviewer deems our responses detailed enough and satisfactory we encourage the reviewer to potentially consider a fresher evaluation of our paper with these responses in context and potentially upgrade their score.

---

> > > > ### Author Response · Authors · 2024-11-24
> > > >
> > > > Dear reviewer,
> > > >
> > > > We are very grateful for your time, effort, and constructive comments. As the end of the rebuttal period is quickly approaching, we would like to have the opportunity to answer any remaining questions or clarify any points. We would like to note that we have followed your suggestion to conduct a detailed analysis of the population embeddings, in both the synthetic letters and biological experiments, strengthening our empirical findings. We also tried to clarify the differences between MFM and existing methods, such as "Learning single-cell perturbation responses using neural optimal transport" and "Deep Generalized Schrödinger Bridge", and how MFM differs from the methods and problem settings considered in these works.
> > > >
> > > > We would be happy to continue to engage on these points or any other additional points that may arise. We again thank the reviewer for their constructive review of our paper. If the reviewer finds that our rebuttal addresses their questions and concerns, we would be grateful if the reviewer would potentially consider a fresh assessment of our work and possibly consider increasing their score.

---

> > > > > ### Comment · Reviewer_kKjh · 2024-11-26
> > > > >
> > > > > The rebuttal addresses many of my questions. I am adjusting my rating 5 --> 6.

---

> > > > > > ### Author Response · Authors · 2024-11-28
> > > > > >
> > > > > > We thank the reviewer for their time and effort in reviewing our work and reading our rebuttal. We are glad to see that our rebuttal and updates to the manuscript have improved the reviewer's evaluation of our work. We are happy to answer any lingering questions or address any new comments that may arise.

---

### Official Review · Reviewer_Y1RT · 2024-11-03

**Soundness:** 2
**Presentation:** 3
**Contribution:** 3
**Rating:** 6
**Confidence:** 4

**Summary:**

The paper introduces an extension to Flow Matching (FM) called Meta Flow Matching (MFM) that efficiently accommodates generalizing to varying base distributions by featurizing the base distribution with a nearest-neighbor graph neural network (GNN).

More specifically, FM is a recent popular framework for generative modeling that works by first matching up a target distribution with a base distribution (e.g., a standard normal distribution, coupled to the target distribution via an independent coupling) and then training a vector field to match the resulting particle vector fields via minimizing the Mean Squared Error (MSE). A natural extension of this is Conditional Flow Matching (CFM) where the vector fields can be conditioned on some covariate, such as a generative prompt for image generation. The authors argue that there are compelling reasons to extend the problem setup to include varying base distributions. First, naturally occurring dynamics such as interacting particles or diffusion equations have dynamics that are explicitly distribution dependent and thus their particle versions cannot be modeled by vector fields that only depend on the particle location alone. Second, problems in molecular biology naturally lend themselves to considering different base distributions, such as predicting the drug response for different baseline cell distributions per patient.

Therefore, the authors suggest expanding the FM formulation with a distributional embedding that captures the base distribution and that is used as input to the vector field neural network. To obtain the embedding, they first calculate a k-nearest-neighbor graph on the base distribution and then embed it with a GNN, leading to the proposed MFM formulation. They show that MFM can be trained in the same manner as FM and show that it outperforms baseline architectures in experimental results on synthetic data (denoising rotated letter profiles) and on real data (mass-cytometry cell profiles on patient-derived cultures under drug treatments).

**Strengths:**

* The manuscript is overall well-written and organized.
* Flow matching is a popular generative modeling framework and is of significant interest to the ML community. I like the authors' motivation of the distribution dependence through the continuity equation and interacting particle systems. Their idea to solve this problem is simple and elegant and deserves exploration.

**Weaknesses:**

* **Empirical results are not fully convincing and need strengthening:** The presentation of the empirical results leave me slightly puzzled, as explained in the following.
    * First of all, the authors consider [ICNN/CellOT](https://www.nature.com/articles/s41592-023-01969-x) as a baseline that can accommodate varying baseline distributions and conditioning. I think the introduction to ICNN presented in in Section 4 is slightly misleading: in lines 348-350, it is sometimes not clear what "the method" refers to. ICNN *can* generalize to new distributions, but the novelty in MFM is to take additional interactions into account.
    * Since ICNN is conceptually the closest competitor to MFM, why did the authors not benchmark their method on the [dataset consider in that manuscript](https://onlinelibrary.wiley.com/doi/epdf/10.1111/exd.12683)? The chosen dataset in Section 5.2 seems well suited for the task, but it leaves me wondering why they swapped out datasets. Did their method not give them the desired results on previously considered datasets?
    * The authors often claim that one method, in particular FM, "fails to fit the train data" (e.g., lines 472-473). This is a strong statement given that FM often surprisingly scores second-best out of the baseline methods. This makes me wonder about the underlying dataset and the employed metrics. At a minimum, I suggest reporting $\mathcal{W}_1$, $\mathcal{W}_2$ and MMD for the following very simple uninformed baselines to account for trivial underlying biological phenomena:
        * The base distribution itself, i.e., distance of the unperturbed patient cells to the perturbed cells without applying any model.
        * The base distribution shifted by a constant vector that is the mean over all perturbed cells (e.g., those in the train dataset)
    * Further, are there ways to better understand the overall quality of the predictions? Are there downstream conclusions of the dataset paper that can or cannot be reproduced by the various predictive algorithms? Can the authors plot some of the predicted distributions just as they did in Figure 3?
    * The generalizability of the results might be compromised by the authors' picking specific patients for the patient holdout setup (see Section C.2). Can the experiment be performed over several splits?
    * In the same vein, I do not understand how the authors arrive at the error estimates for their experimental results if only one split is considered. I am afraid these estimates could be vastly underestimating the actual uncertainty. This is important since MFM sometimes only shows a small edge compared to FM.
* **Independent matching may be supoptimal for all considered methods:** [Tong et al (2023)](https://arxiv.org/abs/2302.00482) propose to train flow matching by coupling base and target distribution via Optimal Transport instead of an independent coupling. This is a simple tweak that could be applied to all considered FM methods, i.e., FM, CFM, and MFM. I am curious if this would improve results across the board or specifically help FM and CFM because those methods currently have no way of accounting for different source distributions. Potentially, this could be a much simpler fix than MFM.
* **(minor)** The text contains quite a few typos. I suggest the authors do another round of proof-reading for the final version.

**Questions:**

I am recapitulating some of the problems outlined in "Weaknesses" above as questions to the authors here, as well as some other things that were not clear to me:

* How are uncertainties on your results calculated? If they are not derived from independent splits/seeds, I would suggest you do that.
* Why did you not benchmark MFM on the [dataset consider in the CellOT manuscript](https://onlinelibrary.wiley.com/doi/epdf/10.1111/exd.12683)?
* How is CFM set up on the patient data set? I assume you condition on the drug, while ignoring the base distribution/patient identity. Is this correct, or are you also feeding the patient identity as condition? If the latter, I would recommend the former.
* Can you add the following uninformed baselines to the experiments in Sections 5.1 and 5.2?
    * The base distribution itself, i.e., distance of the unperturbed patient cells to the perturbed cells without applying any model.
    * The base distribution shifted by a constant vector that is the mean over all perturbed cells (e.g., those in the train dataset)
* Are there ways to better understand the overall quality of the predictions? Are there downstream conclusions of the paper that introduced that can or cannot be reproduced by the various predictive algorithms? Could you plot some of the predicted distributions just as they did in Figure 3?
* I am surprised by the dynamic range of the Wasserstein distances vs MMD. For example, in Table 2, ICNN on the patient holdout achieves MMD of 74.00 and $\mathcal{W}_2$ of 4.681 vs MFM (k=100) with 8.96 and 4.269, respectively. Is there any intuition about this discrepancy in dynamic range?
* How is the $r^2$ calculated here? The model does not actually predict an output for a single target cell, but a distribution instead, so I am confused as to how this is done.
* Would you consider running experiments with Optimal Transport matching instead of independent coupling as considered in [Tong et al (2023)](https://arxiv.org/abs/2302.00482)?

---

> ### Author Response · Authors · 2024-11-21
> **(1/3)**
>
> We thank the reviewer for their time in effort in constructing this in-depth review of or paper, offering meaningful suggestions, and insightful questions. We are happy to see that the reviewer found our work well motivated, that our "idea to solve this problem is simple and elegant", and found our manuscript "overall well-written and organized". Below we address the clarifying questions and suggestions brought up by the reviewer.
>
> >First of all, the authors consider ICNN/CellOT as a baseline that can accommodate varying baseline distributions and conditioning. I think the introduction to ICNN presented in Section 4 is slightly misleading: in lines 348-350, it is sometimes not clear what "the method" refers to. ICNN can generalize to new distributions, but the novelty in MFM is to take additional interactions into account.
>
> Thank you for pointing this out. What we refer to as the "method" here is "CellOT", while the base architecture of CellOT is an ICNN. Throughout the paper and results, we use ICNN to denote the CellOT baseline (see lines 480-483). We also clarify that the CellOT/ICNN model can generalize to new cells, but is not designed to generalize to unseen distributions. We have updated lines 348-350 to clarify this in the text. Further, we reinforce this through our empirical results, where akin to FM, CellOT (ICNN) struggles to generalize across unseen distribution/populations relative to MFM.
>
> We wish to clarify that the novelty of MFM is 2 fold: (1) we can train a generative model to condition on entire distributions through the use of a population embedding model which learns embeddings of entire distributions/populations, and (2) MFM can take into account additional interactions of cells/particles. This differs from CellOT which does not condition on entire distributions, does not learn embeddings for entire populations, and does not take into account interactions of cells. We also refer to the paragraph starting on line 57 where we discuss the difference between MFM and existing methods (including CellOT). On line 63-66, we state one of the limitations of these existing methods in that they are restricted to operating on a single measure.
>
> >Since ICNN is conceptually the closest competitor to MFM, why did the authors not benchmark their method on the dataset consider in that manuscript? The chosen dataset in Section 5.2 seems well suited for the task, but it leaves me wondering why they swapped out datasets. Did their method not give them the desired results on previously considered datasets?
>
> We thank the reviewer for bringing up this important related work. The CellOT datasets are not suitable for the MFM task/objective as there is no opportunity to generalize between initial distributions. The datasets in CellOT have a single control (initial) distribution with multiple treatment conditions. In contrast, the organoid drug-screen dataset considered in our work contains many distribution/population pairs $(p_0, p_1)$ -- specifically, after pre-processing, we have 927 control and treated distribution pairs $(p_0, p_1)$. The problem addressed by MFM is analogous to distributional regression, where each data point is an entire distribution, with the task to generalize across unseen distributions. This problem is ill-posed if there are not sufficient quantity of pairs $(p_0, p_1)$, hence rendering the datasets from CellOT unsuitable.
>
> >I suggest reporting $\mathcal{W}_1$, $\mathcal{W}_2$, and MMD for the following very simple uninformed baselines to account for trivial underlying biological phenomena:
>     - The base distribution itself, i.e., distance of the unperturbed patient cells to the perturbed cells without applying any model.
>     - The base distribution shifted by a constant vector that is the mean over all perturbed cells (e.g., those in the train dataset)
>
> This is a great suggestion. We have added the two additional baselines suggested by the reviewer. Namely, we denote $d(p_0, p_1)$ for comparison of the unperturbed source distribution $p_0$ with the base target distribution $p_1$. We use $d(p_0 - \mu_0 + \tilde{\mu}_1, p_1)$ to denote a comparison between the base target distribution with the source distribution shifted by the difference in means between the perturbed cells $\tilde{\mu_1}$ and the unperturbed cells $\mu_0$. We compute $\tilde{\mu_1}$ by taking the average of all $p_1$ population means in the training set. In Table 2 and Table 3, we see that all models perform better than these simple baselines.
>
> >The authors often claim that one method, in particular FM, "fails to fit the train data" (e.g., lines 472-473). This is a strong statement given that FM often surprisingly scores second-best out of the baseline methods.
>
> We thank the reviewer for pointing this. We agree this is perhaps too strong of a statement and have removed this statement from the text.

---

> ### Author Response · Authors · 2024-11-21
> **(2/3)**
>
> >Further, are there ways to better understand the overall quality of the predictions? Are there downstream conclusions of the dataset paper that can or cannot be reproduced by the various predictive algorithms? Could you plot some of the predicted distributions just as they did in Figure 3?
>
> This is an excellent question. To draw insights towards downstream conclusions on the organoid drug-screen dataset, we compute and plot 2-D projections of embeddings outputted from the population embedding model $\varphi(p_0; \theta)$ (see Figure 7 in Appendix F.1). We observe that we recover grouping of patient populations which are generally consistent with the findings of (Zapatero et al. 2023). For instance, populations from patients 99 and 109 (both chemorefractory) are grouped together, and populations from patients 23 and 27 (both chemosensitive) show lower pairwise distances. We discuss these results on lines 1200-1205 in Appendix F.1.
>
> >The generalizability of the results might be compromised by the authors' picking specific patients for the patient holdout setup (see Section C.2). Can the experiment be performed over several splits?
>
> We thank the reviewer for bringing up this point. We agree and have updated the results for the patient split to include evaluation across left-out populations from 3 different patients. We report the mean and standard deviation of performance metrics across these 3 splits (please see updated Table 3). We observe that MFM outperforms baselines in this setting while also exhibiting robustness across splits.
>
> >In the same vein, I do not understand how the authors arrive at the error estimates for their experimental results if only one split is considered. I am afraid these estimates could be vastly underestimating the actual uncertainty. This is important since MFM sometimes only shows a small edge compared to FM.
>
> For the replica split (Table 2) uncertainties over 3 independent model seeds (please refer to the last sentence of the paragraph on lines 365-373). Specifically, we use this to show that models are robust to changes in model initialization. Following the helpful suggestion of the reviewer, for the patients split (Table 3) we report mean and standard deviation over 3 different left-out patient splits to show performance robustness across different patients.
>
> >**Independent matching may be suboptimal for all considered methods:** Tong et al (2023) propose to train flow matching by coupling base and target distribution via Optimal Transport instead of an independent coupling. This is a simple tweak that could be applied to all considered FM methods, i.e., FM, CFM, and MFM. I am curious if this would improve results across the board or specifically help FM and CFM because those methods currently have no way of accounting for different source distributions. Potentially, this could be a much simpler fix than MFM.
>
> We thank the reviewer for their valuable point. We have added experiments that incorporate optimal transport (OT) couplings between source and target distributions for the synthetic letters dataset (Table 4) and for the organoid drug-screen dataset (Table 2).
>
> We observe that using OT couplings generally improves performance across all methods, with MFM or MFM-OT still yielding the best performance on the left-out test data. We note that adding minibatch OT couplings alone does not provide a way of accounting for generalizing across source distributions.
>
> We thank the reviewer again for this comment. We believe these additional experiments help clarify that it is not the use of optimal transport that allows generalization across initial populations, but the population embedding in MFM.
>
> >How is CFM set up on the patient data set? I assume you condition on the drug, while ignoring the base distribution/patient identity. Is this correct, or are you also feeding the patient identity as condition? If the latter, I would recommend the former.
>
> In short, we report results for both. Namely, all the methods are conditioned on the treatment, then FM ignores the base distribution/patient identity (the former in your question), but CFM (as we indicate via CGFM) additionally conditions on the population identity conditions (the latter in your question). Each patient has numerous $(p_0, p_1)$ population pairs. We have updated lines 483-485 and Appendix D.2 to clarify this. For CGFM, we condition on these population identities as known conditions. Population identity conditions for CGFM are represented as one-hot vectors, hence the model cannot generalize to unseen population identities. We state this on lines 426-427.

---

> > ### Author Response · Authors · 2024-11-21
> > **(3/3)**
> >
> > >I am surprised by the dynamic range of the Wasserstein distances vs MMD. For example, in Table 2, ICNN on the patient holdout achieves MMD of 74.00 and $\mathcal{W}_2$ of 4.681 vs MFM (k=100) with 8.96 and 4.269, respectively. Is there any intuition about this discrepancy in dynamic range?
> >
> > We thank the reviewer for pointing to this. We note that MMD measures the local discrepancy while Wasserstein distances measures global distance variations and that we report MMD measures times $10^{-3}$ (as specified in the table header). It is also more sensitive to small differences in density. It is therefore possible for MMD to be large while Wasserstein distance to be only mildly affected when there are local density variations. It should be noted that while we used unweighted Euclidean distance as the cost for calculating the Wasserstein distance, for our MMD metric, we followed past literature setting the bandwidth hyperparameter $\sigma$ used in MMD to cover different scales. To clarify this we have added Appendix D.3. where we describe and explicitly define these metrics.
> >
> > >How is the $r^2$ calculated here? The model does not actually predict an output for a single target cell, but a distribution instead, so I am confused as to how this is done.
> >
> > The $r^2$ is calculated as the correlation of the correlation of gene expression within the sample distribution between ground truth and the generated sample. For a full mathematical description see Appendix D.3 where we have added further description of metrics. It reflects that genes are expressed at a similar level between cells in the ground truth should also be similar in generated samples. This is a somewhat strange evaluation for a distribution, as it calculates the correlation of correlation, but is a typical evaluation of single-cell prediction methods (Bunne et al. 2023) as it reflects downstream tasks such as gene set enrichment analysis.
> >
> > >(minor) The text contains quite a few typos. I suggest the authors do another round of proof-reading for the final version.
> >
> > We thank the reviewer for eluding to this. We have gone through the text and fixed typos that were originally present in the manuscript.
> >
> > We again thank the reviewer for their very valuable feedback, insightful questions, and time spent in reviewing our work. We believe that through incorporating all the meaningful suggestions brought up by the reviewer, we have improved the overall quality of our work and strengthened our empirical results. We hope that our rebuttal fully addresses the reviewer's concerns, and we kindly ask the reviewer to consider increasing their score if they are satisfied with our response. We are happy to answer any further questions that may arise.

---

> > > ### Author Response · Authors · 2024-11-24
> > >
> > > Dear reviewer,
> > >
> > > We are very grateful for your time, constructive comments, and insightful questions. As the end of the rebuttal period is fast approaching, we would like to have the opportunity to answer any remaining questions or concerns. We would like to note that in our rebuttal we followed your great suggestions and included several new experiments to strengthen our empirical results. We also tried to highlight in both our global response and the rebuttal response the differences between the approach and problem setting in MFM compared to the method and problem setting considered in Bunne et al. 2023.
> > >
> > > We would be happy to engage in any further discussion on these points or answer any additional questions that the reviewer finds important. We thank the reviewer again for their time and effort. If the reviewer finds our rebuttal and new experimental findings satisfactory, we would also appreciate it if the reviewer could potentially consider a fresher evaluation of our paper and kindly ask the reviewer to possibly consider improving their score.

---

> > > > ### Comment · Reviewer_Y1RT · 2024-11-27
> > > > **Thank you for your reponse!**
> > > >
> > > > Thank you for addressing many of the points that I raised! I still have a few comments:
> > > >
> > > > Regarding the organoid dataset, could you clarify whether there are actually multiple replicates for the same patient-drug pair, or do you use replicate interchangeably with drug treatments (per patient) here? I'm getting a bit confused looking at the new Figure 7 on why there are multiple dots per patient. Are you also plotting the treated populations, or multiple replicates for the control population? If there are multiple ones for the control, how do you handle that in the prediction setup?
> > > >
> > > >
> > > > > Ask for downstream conclusions, addition of Figure 7
> > > >
> > > > Thanks for adding this! I appreciate that this helps to understand the quality of the embedding. But is there a similar way to understand the quality of the predictions? I.e., are there simple binary conclusions one would draw from the data (i.e., a specific subset of drugs is effective for a specific subset of patients) that you would be able to answer from the predictions?
> > > >
> > > >
> > > > > $r^2$ explanation
> > > >
> > > > Frankly, I still don't fully understand how it is calculated. Would it be possible to add to the description the specific handling of the dimensionality involved? I assume one correlation is applied on genes and the other one on cells, but I'm not sure which one is which.

---

> > > > > ### Comment · Reviewer_Y1RT · 2024-11-27
> > > > >
> > > > > > $d(p_0, p_1)$ and $d(p_0 - \mu_0 + \tilde \mu_1, p_1)$ baselines
> > > > >
> > > > > Thanks for adding these! I'm surprised to see the shifted mean baseline perform much more poorly than the simple control population baseline on the biological dataset. This suggests to me that 0 is a better mean estimate than the mean of the training samples. Do you have an idea of what's going on here? Did you calculate $\mu_1$ based on all populations, or only those that share the same source population? For the replicate split, the latter would be more appropriate I think.

---

> > > > > > ### Author Response · Authors · 2024-11-28
> > > > > > **Reply (1/2)**
> > > > > >
> > > > > > Thank you for your time and effort in giving constructive and meaningful feedback for our work!
> > > > > >
> > > > > > >could you clarify whether there are actually multiple replicates for the same patient-drug pair, or do you use replicate interchangeably with drug treatments (per patient) here?
> > > > > >
> > > > > > This is an excellent question. For each patient, **there are multiple replicates** for the same patient-drug pair -- i.e. multiple $(p_0, p_1)$ pairs. There are approximately three technical replicates for every experimental setting. There is slight variation due to experimental variability (particularly of antibody effectiveness leading to reduced yield in some experiments).
> > > > > >
> > > > > > >I'm getting a bit confused looking at the new Figure 7 on why there are multiple dots per patient. Are you also plotting the treated populations, or multiple replicates for the control population?
> > > > > >
> > > > > > In Figure 7, we are only plotting the population embeddings for the replicated control populations $p_0$. Following from our clarification in the previous point, what you observe in Figure 7 as the multiple dots per patient is due to the numerous control populations for each patient. This is the central artifact of this dataset that lets us approach the problem of learning embeddings of entire populations and generalizing to unseen $(p_0, p_1)$ (given unseen $p_0$ predict treatment response $\hat{p}_1$).
> > > > > >
> > > > > > We note that this somewhat complicated setting of controls was done in the previous paper to control for batch effects. Specifically, each treatment has a matched control on the same 96-well plate to match the experimental conditions as closely as possible. Due to the size of the dataset, roughly 4 treatments can fit on a single plate. These will all use the same matched control. However, since the drug screen is so large, experiments have to be done on multiple plates. This means that there are a number of separate controls for each patient depending on the exact setup of plates and treatments, and reruns of failed data capture. We hope this clarifies the reasoning behind the control setup of the prior work.
> > > > > >
> > > > > > >If there are multiple ones for the control, how do you handle that in the prediction setup?
> > > > > >
> > > > > > During prediction, we observe a new (unseen) control population $p_0$, and we ask the question of how well we can predict $p_1$ (for treatments seen during training). Specifically, MFM learns to represent entire populations $p_0$. At test-time, the vector field model predicts conditional population dynamics, conditioned on population embeddings $\varphi(p_0; \theta)$ for an *unseen* test control population $p_0$ and additionally conditioned on a known treatment, to recover $p_1$.
> > > > > >
> > > > > > Given that population pairs $(p_0, p_1)$ are coupled, we are able to validate and evaluate how well we predict $p_1$ given said treatment for an *unseen* $p_0$. Regarding Figure 7, we are not plotting embeddings of the predicted and treated $p_1$ populations, we just plot the learned population embeddings $p_0$ across the entire dataset to demonstrate our model can recover known biological artifacts which are also found by (Zapatero et al. 2023).
> > > > > >
> > > > > > >Is there a similar way to understand the quality of the predictions? I.e., are there simple binary conclusions one would draw from the data (i.e., a specific subset of drugs is effective for a specific subset of patients) that you would be able to answer from the predictions?
> > > > > >
> > > > > > Yes, we similarly provide embedding plots of the ground truth target and predicted populations separately in Figure 8. We can see here that for all three test patients, the general structure is preserved, with the treatment, Oxaliplatin (Green), being the furthest away both for the target and ground truth datasets. This is because (as shown in the original paper by Zapatero et al. 2023) Oxaliplatin has a large effect on these cancer cells for this subset of patient-derived organoids (PDOs), as we can see from the plots this is more pronounced for PDO 21 and 27 than for PDO 75. This reflects the conclusions drawn from Figure 4 in the original dataset paper (Zapatero et al. 2023). In this way, we are able to draw similar conclusions from the predicted distributions as from the data. We add additional details on this experiment in Appendix F.2.

---

> > > > > > > ### Author Response · Authors · 2024-11-28
> > > > > > > **Reply (2/2)**
> > > > > > >
> > > > > > > >I still don't fully understand how $r^2$ is calculated. Would it be possible to add to the description the specific handling of the dimensionality involved? I assume one correlation is applied on genes and the other one on cells, but I'm not sure which one is which.
> > > > > > >
> > > > > > > To make this clearer, in Appendix D.3 lines 884-900 we have further elaborated on how the $r^2$ metric is computed. We follow the implementation done by Bunne et al. 2023. In Appendix D.3 we have added formal definitions of the quantities being computed and the handling of the dimensionality. We are happy to further elaborate and answer any remaining questions regarding the computation of the $r^2$ metric.
> > > > > > >
> > > > > > > >I'm surprised to see the shifted mean baseline perform much more poorly than the simple control population baseline on the biological dataset. This suggests to me that 0 is a better mean estimate than the mean of the training samples. Do you have an idea of what's going on here?
> > > > > > >
> > > > > > > This behaviour is expected from this datatype since it is a mixture of two cell types (cancer cells and fibroblasts) where the proportion between the two populations changes drastically. This is because one of the major treatment effects is to kill the cancer cells (reducing their proportion relative to the fibroblasts). This means that the shift $\tilde \mu_1 - \mu_0$ is a transformation towards a fibroblast state (because $p_1$ has relatively more fibroblasts as the cancer cells have been killed by the treatment). This creates very weird predictions where it looks like the cancer cells are turning into fibroblast cells (which is not possible biologically). This makes the mean shift estimate quite poor here as it creates cells that are very unlikely to appear in the dataset. We can see this effect is particularly pronounced in the relative change of the MMD metric between the null shift and the mean shift baselines, which measure local differences in densities.
> > > > > > >
> > > > > > > >Did you calculate $\mu_1$ based on all populations, or only those that share the same source population? For the replicate split, the latter would be more appropriate I think.
> > > > > > >
> > > > > > > To further clarify this, we have added an additional baseline $d(p_0 - \mu_0 + \mu_1, p_1)$ that uses the *individual* population means $\mu_1$ of treated populations $p_1$ for each coupled pair $(p_0, p_1)$. We report this in updated Tables 5, 6, 7. Here, $d(p_0 - \mu_0 + \mu_1, p_1)$ differs from $d(p_0 - \mu_0 + \tilde{\mu}_1, p_1)$ as $\tilde{\mu}_1$ is estimated as the average mean (or mean of means) of the treated populations present in the training data. Specifically, we compute $\mu_1$ for each individual treated population $p_1$ in the train data. Then, $\tilde{\mu}_1$ is estimated as $\tilde{\mu}_1 = 1/N \sum_i^N \mu^{(i)}_1$, where $N$ is the number of $p_1$ populations in the train data. The $d(p_0 - \mu_0 + \tilde{\mu}_1, p_1)$ baseline shows that one cannot trivially estimate some constant shift $\tilde{\mu}_1$ to predict $p_1$. In regards to $d(p_0 - \mu_0 + \mu_1, p_1)$ (which uses the true $\mu_1$'s for each $(p_0, p_1)$ pair), we observe that indeed this baseline does perform better than the trivial baseline $d(p_0, p_1)$, as one might expect. Note, we report $d(p_0 - \mu_0 + \mu_1, p_1)$ only on the train data, since in practice you do not have access to $p_1$ at test-time, so this is not a practical baseline to consider in the test condition.
> > > > > > >
> > > > > > > We once again thank the reviewer for their valuable comments and questions. We hope this discussion addresses the reviewer's remaining points. We are more than happy to keep clarifying and addressing any salient points that may remain. We believe that through this discussion, we have improved the overall quality of our work and strengthened our empirical findings. We kindly ask that if the reviewer views our responses and additions to the manuscript as satisfactory, to consider increasing their rating of our paper.

---

> > > > > > > > ### Author Response · Authors · 2024-12-01
> > > > > > > >
> > > > > > > > Dear reviewer,
> > > > > > > >
> > > > > > > > We thank you again for your time and effort in providing a constructive and insightful review of our work. As the end of the discussion period is fast approaching, we hope we have addressed all your questions and concerns. If you have any more comments, we would be happy to engage in further discussion.

---

> > > > > > > > > ### Comment · Reviewer_Y1RT · 2024-12-01
> > > > > > > > >
> > > > > > > > > Thanks again for the clarifications and additional analyses! I'm adjusting my rating from 5 to 6.
> > > > > > > > >
> > > > > > > > > I would still recommend a few edits regarding the last points we discussed:
> > > > > > > > >
> > > > > > > > > > Figure 8
> > > > > > > > >
> > > > > > > > > If the text is correct, then I think the axis labels should say PCA instead of UMAP.
> > > > > > > > >
> > > > > > > > > > $r^2$
> > > > > > > > >
> > > > > > > > > Thank you for clarifying this a bit more in the paper. If I understand it correctly, you are calculating the correlation matrix over genes within each population and then calculating the overall correlation between the genes, comparing the two samples. I would take another look at the math: for example, if it's a matrix you calculate in eq. (39), you would want to add a transpose somewhere. I also believe that the notation $X_i$ is overloaded, once referring to an observation and then to a gene.

---

> > > > > > > > > > ### Author Response · Authors · 2024-12-02
> > > > > > > > > >
> > > > > > > > > > Thank you for your constructive feedback!
> > > > > > > > > >
> > > > > > > > > > >If the text is correct, then I think the axis labels should say PCA instead of UMAP.
> > > > > > > > > >
> > > > > > > > > > Thank you for pointing this out, this is correct, the axis labels should read PCA instead of UMAP. We have fixed this typo in the manuscript.
> > > > > > > > > >
> > > > > > > > > > >if it's a matrix you calculate in eq. (39), you would want to add a transpose somewhere. I also believe that the notation $X_i$ is overloaded, once referring to an observation and then to a gene.
> > > > > > > > > >
> > > > > > > > > > Thank you for identifying this. We have adjusted this in the text.
> > > > > > > > > >
> > > > > > > > > > We will keep improving the presentation for the next version of the manuscript. We thank the reviewer for their meaningful suggestions that have helped improve the quality of the paper and are glad to see that our rebuttal and updates have improved the reviewer's evaluation of our work. We are happy to answer any new questions that may arise.

---

### Official Review · Reviewer_Kj6P · 2024-11-04

**Soundness:** 2
**Presentation:** 1
**Contribution:** 3
**Rating:** 5
**Confidence:** 2

**Summary:**

This article deals with the problem of learning many-body dynamics via flow matching. The main objective is to model the interaction between components (e.g. particles) rather than modeling the entities separately. The resulting methodology allows to correctly model the evolution of distributions from the time evolution of observations of sample populations.

**Strengths:**

-- The problem is known to have plagued science for a long time. Therefore, new approaches and solutions are important contributions.
-- The idea of modeling the fields in the Wasserstein setting is intriguing, and intuitively meaningful.
-- The authors relate their approach with mean-field limits and diffusion processes. This is certainly something of interest.

**Weaknesses:**

-- First and foremost, the article is written in a very confusing way. It seems that the narrative is scattered, and this really makes it difficult to follow.
-- The problem seems to boil down to embedding populations using graph neural networks and then flow matching in Wasserstein manifolds.
-- It is unclear what a Wasserstein manifold is to start with. It clearly should not be a finite dimensional manifold, as depicted in their Figure 2. This is an infinite dimensional manifold, but the atlases are not clearly specified. For instance, a Banach manifold is a well understood object. In this case, the authors do not bother to introduce what they are talking about. P_2(X) is not generally even a Banach manifold. My understanding is that P_2(X) has a geometric structure only in some relatively special cases (Otto calculus), and usually people talk about metric over P_2(X), and the notion of curvature in Wasserstein spaces is often not the curvature of a manifold in the traditional sense of differential geometry. How is the tangent space defined in this context? In section 2.3 it seems that the author suggest that using the Wasserstein setup is rather important. As a consequence, I think it should not be left to the interpretation of the reader, but it should be clearly explained (at least defined).
-- In general, there is a terrible lack of definitions. Even the theorem, which should be a formalization of the work, is unclear because the definitions are either absent, or scattered throughout the article. At least an appendix with all definitions should be added.
-- The results do not seem overall particularly good. The table included in the text seems to show good results, but those in the appendix seem to show that the model is not particularly outperforming. One might not care much about this if the paper were a very well written piece of theoretical work, but as explained above it does not seem to be that case either.

**Questions:**

All weaknesses as stated above are my main questions.

---

> ### Author Response · Authors · 2024-11-21
>
> We thank the reviewer for their detailed feedback and constructive comments, which gave us an opportunity to improve the clarity of our work significantly. We are pleased to see that the reviewer found our work intriguing and intuitively meaningful. We now address key clarification questions raised by the reviewer.
>
> >First and foremost, the article is written in a very confusing way.
>
> Thank you for bringing this up, we incorporated the changes you suggested into the manuscript.
>
> >It is unclear what a Wasserstein manifold is to start with.
>
> Thank you for the suggestion, we updated the manuscript correspondingly (see Appendix A, lines 157-161). Note that the Wasserstein manifold is "the Riemannian interpretation of the Wasserstein distance developed by Otto" (Ambrosio et al., page 168) and is a well-defined and a well-studied abstract formalism. Indeed, the metric of the Wasserstein manifold is the Wasserstein metric, and the tangent space is defined by the gradient flows as we discuss in lines 157-161. Figure 2 is just an illustration rather than a precise depiction of the Wasserstein manifold. Note that we clearly state in the caption and describe in the text that this is the space of distributions $\mathcal{P}_2(X)$. We are confident that under no circumstances the reader might think that a sphere serves as a precise depiction of the infinite-dimensional space of distributions.
>
> >At least an appendix with all definitions should be added.
>
> Thank you for your suggestion, we added Appendix A including the definitions of the concepts we are using in the paper. We believe this increases the clarity of our method.
>
> >The table included in the text seems to show good results, but those in the appendix seem to show that the model is not particularly outperforming.
>
> Thank you for raising this concern. We respectfully disagree that the appendix shows that the model is not particularly outperforming. We believe this misunderstanding maybe due to the inclusion of evaluation on the train data for reference in the appendix. We note that we primarily care about model performance on the test data and not the train data in this work. To clarify this, we have added bolding to the best performing method in the appendix tables to further illustrate that our model outperforms baseline methods. We hope this clarifies that MFM outperforms baselines in all settings on the test data.
>
> We thank the reviewer for their valuable feedback and great questions. We believe through incorporating the feedback provided by the reviewer, we have improved the clarity and quality of our manuscript. We hope that our rebuttal fully addresses all the salient points raised by the reviewer and we kindly ask the reviewer to potentially upgrade their score if the reviewer is satisfied with our responses and updated manuscript. We are also more than happy to answer any further questions that arise.

---

> > ### Author Response · Authors · 2024-11-24
> >
> > Dear reviewer,
> >
> > We are very thankful for your time and insightful suggestions. As the end of the rebuttal period is fast approaching we would like to have the opportunity to answer any remaining questions. We would like to note that in our rebuttal we followed your great suggestions and added clarifications in the text to improve clarity and reduce confusion. To note, we added Appendix A, which includes definitions of the concepts we use in this work. We also clarified the presentation of our results and our depiction of the Wasserstein manifold in the rebuttal.
> >
> > We would be happy to engage in any further discussion on these points or any additional points that the reviewer may find important, please let us know! We thank the reviewer again for their time and if the reviewer finds our rebuttal and additions to the manuscript satisfactory, we would also appreciate it if the reviewer could potentially consider revising their assessment of our paper and improving their score.

---

> > > ### Comment · Reviewer_Kj6P · 2024-11-28
> > >
> > > Few more points.
> > >
> > > Overall, including the appendix seems to have helped, but still there are some vague things.
> > >
> > > -- At page 168, referenced in the answer by the authors, it does not seem that the phrase states that "a Wasserstein manifold is...". Actually, it seems that the word "manifold" appears only four times in the book (at least the version I have). So, the concept seems to have been introduced in this article mistakingly, as an equivalent definition to "Wasserstein space" which seems to appear several times in the book by Ambrosio et al. I suggest to add the definition clearly in the appendix, saying something like "a Wasserstein space with the bundle of tangent spaces defined by ... is called Wasserstein manifold". It seems absurd that the concept that is at the base of the whole work is not defined anywhere explicitly.
> > >
> > > -- Once the tangent space becomes an L2 space, some problems might arise. For instance, the matching now is a sort of average. Errors might be small "on average", but might blow up in certain regions that have not been explored. This is not considered in the article. Is there a reason to assume that it does not give rise to issues?
> > >
> > > -- I have now noticed that some of the errors were train errors, as pointed out. I am unsure why these are included, since it is not usually a metric of particular interest, but this clarifies my previous comment. Is there a reason why OT does not seem to help the other models? Is there a theoretical reason, or just lack of hyperparameter fine-tuning?

---

> > > > ### Author Response · Authors · 2024-11-28
> > > >
> > > > Thank you for your time and effort in reading our rebuttal and for your helpful feedback. Please, note that we can no longer update the manuscript on Openreview. Hence, all the introduced changes will appear in the camera-ready version of the manuscript.
> > > >
> > > > >I suggest to add the definition clearly in the appendix, saying something like "a Wasserstein space with the bundle of tangent spaces defined by ... is called Wasserstein manifold"
> > > >
> > > > Thank you for your suggestion, it significantly improves the clarity of our presentation! We have added the corresponding definition to the paper and stated that by the Wasserstein manifold, throughout the paper, we simply mean the Riemannian interpretation of the Wasserstein space.
> > > >
> > > > >Once the tangent space becomes an L2 space, some problems might arise. Is there a reason to assume that it does not give rise to issues?
> > > >
> > > > Indeed, the optimized objective is an average over the marginals in the state space, which is a common thing in the literature (Ho et al, 2020, Lipman et al, 2023). Note that no “exploration” actually takes place since we are not optimizing over the marginals but rather learning the predefined dynamics. To our knowledge this does not lead to any issues with blowing up errors since it requires the simulation within the training distribution rather than out-of-distribution generalization.
> > > >
> > > > > Is there a reason why OT does not seem to help the other models?
> > > >
> > > > In theory, the optimal transport coupling of individual marginals should not significantly affect the predictions because our goal is to generate the correct distribution, e.g. the correct population of cells. Also, note that the metrics we consider do not depend on the coupling between the initial distribution $p_0$ and the final $p_1$. In practice, we observe that it leads to marginal improvements, likely due to the simpler form of the vector fields that the model has to approximate. However, note that although the improvements are small, all methods, FM, CGFM, and MFM, do exhibit improvement when using OT relative to their non-OT counterpart (Table 2 in the main text). In the letters experiment (Table 4 in the Appendix), the lack of improvement when using OT is likely because the joint distribution $\pi(x_0,x_1) = \mathcal{N}(x_0|x_1,\sigma_t)p_1(x_1)$ corresponds to the samples from the "sharp" silhouette $p_1(x_1)$ with added normal noise $\mathcal{N}(x_0|x_1,\sigma_t)$, which is much closer to OT than independent coupling $\pi(x_0,x_1) = p_0(x_0)p_1(x_1)$.
> > > >
> > > > We again thank the reviewer for their constructive comments and questions, and for engaging in fruitful discussion. We believe through implementing the reviewer's helpful suggestions and through this discussion, the quality of our manuscript has improved. We kindly ask, that if the reviewer deems our response satisfactory, to possibly consider improving their rating of our paper. We are more than happy to further engage in discussion and answer any salient points that the reviewer may have.
> > > >
> > > > Ho, Jonathan, et al. "Denoising diffusion probabilistic models." Advances in neural information processing systems, (2020).
> > > >
> > > > Lipman, Yaron, et al. "Flow matching for generative modeling." International conference on learning representations, (2023).

---

> > > > > ### Author Response · Authors · 2024-12-01
> > > > >
> > > > > Dear reviewer,
> > > > >
> > > > > Thank you for your time and effort in reviewing our work and for providing helpful feedback to improve our manuscript. As the end of the discussion period is fast approaching, we hope we have addressed all your questions and concerns. If you have any more salient points or questions, we would be happy to engage in further discussion.

---

### Official Review · Reviewer_jcAc · 2024-11-05

**Soundness:** 3
**Presentation:** 4
**Contribution:** 4
**Rating:** 8
**Confidence:** 4

**Summary:**

This paper introduces Meta Flow Matching (MFM), a novel approach for modeling the evolution of systems consisting of interacting samples/populations. Unlike previous flow-based models, MFM can generalize to unseen initial populations by using a Graph Neural Network to embed the population and amortizing the flow matching model over these embeddings  The authors demonstrate MFM's effectiveness on an interesting synthetic letter denoising task and a large-scale single-cell organoid screening dataset, showing its ability to predict patient-specific responses to treatments.

**Strengths:**

- Novel approach: The paper proposes a new method for integrating vector fields using the full of probability densities - allowing the modeling of the evolution of entire distributions of samples (which is important for many biological and physical processes). I also believe this amortization effect is especially helpful in small-sample regimes.
- Theoretical foundation: The paper provides a solid theoretical basis for the proposed method, including the connections to existing approaches like conditional generative flow matching.
- Readability: The paper is *very* well written and easy to read and understand (although it's covering a complex topic)

**Weaknesses:**

- Honestly, no weaknesses come to my eye. I've been working on Flows for a quite some time now, and have to say this is a good and well-executed idea.
-  Figure 1 (c) is not that helpful in my opinion.

Typos:
- "a a standard" (l420)
- "of of model" (l464)

**Questions:**

- In Eq.17, shouldn't the condition `c` be part of `\phi` or `v_t` as well?
- I am wondering in what cases I should *not* use MFM to model the change of a distribution using a flow? I can see that in Exp. 5.1 and 5.2, the samples have strong dependencies (e.g., to create the silhouettes), but in case there are no dependencies, would MFM also model this well?

---

> ### Author Response · Authors · 2024-11-21
>
> We thank the reviewer for their time and positive appraisal of our work. We are thrilled that the reviewer viewed our work to be "very well written and easy to understand" and "provides a solid theoretical basis for the proposed method".  We now provide responses to the main questions raised by the reviewer.
>
> > Figure 1.c is not that helpful in my opinion.
>
> We are happy to clarify and amend the visual illustrations in our manuscript. In regards to Figure 1, we have updated the caption to try and clarify the advantages of MFM and how it differs from current approaches. We are open to suggestions on how to improve the visual depiction in this figure to further reinforce the ideas in our work. In regards to Figure 2-c, in the general response, we provide a brief clarification regarding our illustration of the Wasserstein manifold and how it is considered in MFM. We are happy to incorporate any suggestions the reviewer may have to further improve and simplify visual depictions of our framework and method.
>
> > In Eq.17, shouldn't the condition $c$ be part of $\phi$ or $v_t$ as well?
>
> We thank the reviewer for pointing to this important detail. In Theorem 1, $c$ corresponds to the *ideal* condition on the population that we’re trying to learn. Eq. 17 defines an objective function used to optimize a $v(\cdot; \varphi(p_0; \theta) \omega)$, where $\varphi(p_0; \theta)$ is a population embedding model that learns to represent the entire initial population, approximating the *ideal* condition. Hence, Eq. 17 defines, in a sense, an *unconditional* MFM objective in so far that it does not contain any known conditions (such as treatments in our biological setting). From here it is trivial to extend Eq. 17 to the *conditional* setting with any amount of known conditions. For instance, we show this in Algorithm 1 and Algorithm 2, where we use $c^i$ to denote a known treatment condition for population $i$. The condition $c^i$ and the embedding of the population play different roles, e.g. the condition $c^i$ carries the information about the treatment, while the embedding has the information about the patient through the population of cells.
>
> > I am wondering in what cases I should not use MFM to model the change of a distribution using a flow?
>
> Excellent question. MFM relies on there existing a learnable relationship between the source and target distributions. If there is no dependence between the initial and target distributions we would not expect MFM to work well. For instance, we do not expect MFM to be anyhow helpful for the classical setting of generative modelling because there is no any dependence between a given empirical distribution (e.g. natural images) and the Gaussian prior.
>
> We thank the reviewer again for their valuable feedback and great questions. We have implemented the reviewer's suggestions and hope that our rebuttal addresses their questions. We are also more than happy to answer any further questions that arise.

---

> > ### Author Response · Authors · 2024-11-24
> >
> > Dear reviewer,
> >
> > We are very appreciative of your time and positive comments. As the end of the rebuttal period is quickly approaching, we would be happy to answer any remaining or additional questions, please let us know! Again, we thank the reviewer for their time and effort in reviewing our paper.

---

> > > ### Comment · Reviewer_jcAc · 2024-11-26
> > > **Comment by reviewer**
> > >
> > > Thanks for taking the time to answer my questions.
> > >
> > > I usually try to build Figure 1 without any formulas to keep it easily understandable and ensure the reader is not wondering what $h,i,\phi, ...$ actually mean, but I guess everyone has different preferences on that.
> > >
> > > > In Eq.17, shouldn't the condition be part of or as well?
> > >
> > > Thanks for clarifying this. I had the code line corresponding to l.281 in mind when I was thinking about Eq.17. But I agree it is fine to introduce this step by step and not everything at once.
> > >
> > > I was also thinking a bit more about when and when not to use MFM. I guess extrapolation is also quite tricky since there isn't a strong bias (the network $v$ will probably show an arbitrary behaviour outside the training domain).

---

> > > > ### Author Response · Authors · 2024-11-28
> > > >
> > > > Thank you for taking the time to read our rebuttal and for your valuable comments.
> > > >
> > > > Regarding Figure 1, we have updated the figure to provide a better and more clear overview of MFM and how it differs from FM. We would be happy to incorporate any additional suggestions that the reviewer may have.
> > > >
> > > > >I was also thinking a bit more about when and when not to use MFM. I guess extrapolation is also quite tricky since there isn't a strong bias (the network $v$ will probably show an arbitrary behaviour outside the training domain).
> > > >
> > > > We leave the out-of-distribution (OOD) generalization abilities of MFM for the future studies. Namely, in the current paper, we show that MFM can learn the representations that generalize within the training distribution, while the OOD generalization should be approached in a broader context of representation learning, e.g. the representation learning for the single cell data or, more generally, representation learning of distributions.

---

> ### Comment · Reviewer_jcAc · 2024-12-03
> **Official Comment by the Reviewer**
>
> Thanks for your rebuttal.
>
> I increased my Presentation and Contribution Score after the rebuttal because the manuscript became better in that sense, and I strongly believe this is an important contribution to learning dynamics. I am sure many people in my research field will build upon this, and I hope the other reviewers will see it similarly.

---

> > ### Author Response · Authors · 2024-12-03
> >
> > We are excited to see that our work was so positively received and that the reviewer strongly believes that our paper provides an important contribution to learning dynamics! We again thank the reviewer for their positive appraisal of our work and for their constructive and insightful feedback on our manuscript.

---

### Author Response · Authors · 2024-11-21
**General Response**

We thank all the reviewers for their thoughtful feedback, constructive questions, and valuable time spent on reviewing our work, which has helped improve our submission.

In this work, we introduced **Meta Flow Matching (MFM)**, a novel framework for learning the dynamic evolution of populations with the objective to learn population dynamics and generalize to unseen populations/distributions. By amortizing the flow model over initial populations -- using a GNN architecture to learn conditional embeddings of entire populations -- **on synthetic and real (biological) experiments we show that MFM can successfully generalize across previously unseen test populations**, relative to state-of-the-art baselines.

We are excited to see the positive feedback provided by the reviewers. To summarize, the reviewers found our work novel (jcAc, kKjh), well written (jcAc, Y1RT), and well motivated and meaningful (Kj6P, Y1RT). In particular, reviewer jcAc outlined that MFM is novel and that our work provides a strong theoretical foundation for our proposed framework; also stating that our method is a "well-executed idea" with no "no weaknesses". Reviewer Kj6P found that MFM addresses a problem that "is known to have plagued science for a long time" and states that our approach to modeling vector fields on the Wasserstein manifold is "intuitive and meaningful". In a similar vein, reviewer Y1RT likes our motivation for modeling dynamics of interacting particles through the use of distribution conditional continuity equations, describing our work as "elegant, deserving of exploration". Lastly, reviewer kKjh outlined that MFM is "clearly novel".

---

> ### Author Response · Authors · 2024-11-21
>
> In the remainder of this general response we address two points: (1) an overview of the new experiments we ran to address shared questions raised in the reviews; (2) an overview of changes to the manuscript to answer clarifying questions. Please refer to the updated manuscript for all changes mentioned in the general response and responses to individual reviewers. *Changes are presented as blue text in the updated manuscript*.
>
> 1. **Experiments:**
>     - Reviewer Y1RT raised questions regarding the empirical experiments and provided meaningful suggestions to strengthen our results. We incorporated all suggestions from reviewer Y1RT, in turn improving the overall quality of our empirical results. Firstly, we added patient split experiments over 3 independent patient splits (Table 3 in the updated manuscript) to show MFM that is robust across different settings with differing left-out patient populations. Secondly, we added experiments with optimal transport (OT) couplings between samples of source and target distributions for a replica split (updated Table 2). This addition helped improve the overall performance of MFM while also increasing the strength of comparison to stronger baselines, i.e. FM-OT, CGFM-OT. Lastly, for experiments on the organoid drug-screen dataset, we added trivial baselines to demonstrate that the models learn meaningful population dynamics beyond trivial biological phenomena.
>     - Reviewer kKjh asked about analyzing the population embeddings to verify that our population embedding model learns valuable representations. We added a section to the appendix to confirm this (Appendix F.1, Figure 7). Through this, we also addressed reviewer Y1RT's question regarding using our method to draw meaningful insights in the biological dataset.
> 2. **Clarifications:** Reviewers had clarifying questions which we addressed through the individual responses, and in some cases adding clarification to the manuscript. Below we outline and clarify some shared questions asked by reviewers.
>     - Reviewer's Y1RT and kKjh asked clarifying questions regarding comparisons with Bunne et al. 2023 [1]. We note that in our work we indeed do compare with the method in [1]. In the individual responses, we clarified how MFM differs from the method in [1] and how datasets in [1] are not suitable for the setting which MFM is addressing. We made brief changes in the text to further clarify these points.
>     - Reviewer Kj6P and jcAc asked clarifying questions pertaining the theoretical ideas presented in our work. Kj6P asked about our use of the Wasserstein manifold and the respective theory surrounding it. To help with understanding and improve clarity, we provided a clarifying explanation in the individual response, briefly clarified some notation in section 2.3, and added formal definitions in Appendix A of the concepts we are using in the paper. We note that our depiction of the Wasserstein manifold in Figure 2 is only an illustration. We use this figure to illustrate the notion that MFM learns to integrate vector fields on $\mathcal{P}_2(X)$, and because of this, can generalize to unseen populations.
>
> We once again thank all the reviewers for their valuable time and insightful feedback. We believe that through the meaningful feedback and suggestions raised by the reviewers, we have improved the clarity, impact, and significance of our work. Through this, we believe that we have addressed all the concerns and questions posed by the reviewers. If the reviewers agree, we hope that the they will consider increasing their scores.
>
> [1] Bunne, Charlotte, et al. "Learning single-cell perturbation responses using neural optimal transport." Nature methods 20.11 (2023): 1759-1768.

---

### Meta-Review · Area_Chair_hFaM · 2024-12-22

**Metareview:**

The paper proposes a 'meta flow matching approach' for modelling a family of distributions, conditioned on the population index i;
instead of previous approaches (conditional generative flow matching) instead of embedding a sequence i the meta-flow matching approach embeds to population (i.e. the set of points) into the conditioning vector for a vector field. This is very neat and mimics, for example, mean field approaches. The paper is well-written. The applicability is shown for small-dimensional distributions, but overall, this is a good work. Some of the concerns are the actual graph neural network design for particular applications and correct comparison to the baselines, but it also has been A.

**Additional Comments On Reviewer Discussion:**

Most of the reviewers agree this is a good work, except for reviewer Kj6P who found the work confusing, but some of his concerns were addressed by the appendix A.

---

### Decision · Program_Chairs · 2025-01-22

Accept (Poster)